# LEGACY: A LIGHTWEIGHT DYNAMIC GRADIENT COMPRESSION STRATEGY FOR DISTRIBUTED DEEP LEARNING

**Mostapha Essoullami, El Houcine Bergou**
College of Computing
Mohammed VI Polytechnic University
Ben Guerir, Morocco

**Aritra Dutta**
School of Data, Mathematical, and Statistical Sciences
Institute of Artificial Intelligence
University of Central Florida, Orlando, USA

## ABSTRACT

Distributed learning has achieved remarkable success in training deep neural networks (DNNs) on large datasets, but the communication bottleneck limits its scalability. Various compression techniques have been proposed to alleviate this limitation; however, they either use fixed parameters throughout training or rely on complex and computationally intensive methods to adapt compression parameters. Instead of the hard-to-tune hyperparameters required by adaptive compressors, this paper investigates the impact of two fundamental factors in DNN training—the layer size of the networks and their training phases—to design a simple yet efficient *dynamic scheduler* for any compressor, guiding the selection of compression parameters. We present a **L**ightweight **E**fficient **G**r**A**dient **C**ompression strateg**Y** or LEGACY, which, in theory, can work with any compression technique to produce a simple dynamic counterpart. We benchmark LEGACY on distributed and federated training, involving seven different DNN architectures, ranging from ResNet, Transformer-XL, to GPT-2, across large and challenging datasets, including ImageNet, WikiText-103, and OpenWebText. On ImageNet-1K, with an equivalent average data volume, LEGACY's dynamic compression strategies improve the Top-1 accuracy of ResNet-50 by $7 - 11\%$ compared to uniform Top-$0.1\%$ compression, while on WikiText-103, the layer-based dynamic strategy reduces the perplexity of Transformer-XL by $\sim 26\%$ relative to the same baseline. In addition, we evaluate LEGACY under constrained and federated settings, and demonstrate that it scales effectively to a 100-worker configuration while maintaining strong accuracy under aggressive compression. We publish code at:https://github.com/LEGACY-compression/LEGACY.

## 1 INTRODUCTION

Distributed learning on multiple computing nodes is widely adopted to achieve optimal training performance for large deep neural networks (DNNs) (You et al., 2018; Wongpanich et al., 2021; Xu et al., 2021a). However, the training requires exchanging gradients between the nodes; the massive volume of the exchanged data creates a communication bottleneck, and different compressed communication techniques (quantization (Dettmers, 2015; Alistarh et al., 2017; Bernstein et al., 2018), sparsification (Aji & Heafield, 2017; Stich et al., 2018; Alistarh et al., 2018; Dutta et al., 2020), low-rank (Vogels et al., 2019), and hybrid (Basu et al., 2019)) are designed to mitigate this problem.

Among these techniques, sparsifiers achieve baseline performance by only sending a small subset of the gradient components. E.g., by communicating only 0.36% of the largest gradient elements of ResNet-50 (He et al., 2016) trained on ImageNet-1K (Deng et al., 2009), Lin et al. (2018) achieves a baseline no compression performance. Nevertheless, almost a decade after its introduction by Aji & Heafield (2017) for gradient compression, there is no clear recipe for choosing $k$ for training different DNN models using the Top-$k$ sparsifier. While Top-$k$ sends fixed data volume in each training iteration, the threshold sparsifier (a.k.a hard-threshold (Strom, 2015; Sahu et al., 2021)) communicates gradient components with absolute magnitude greater than a threshold, $\lambda \geq 0$. It sets anything less than $\lambda$ to zero. This allows the threshold sparsifier to send a variable amount of data in each iteration and has a better convergence guarantee (Sahu et al., 2021). One can view the threshold sparsifier as a *simple adaptive counterpart of Top-$k$*, since it sends variable amounts of data in each training iteration. However, the same question persists—how to tune the threshold, $\lambda$, in practice?

Not only the sparsifiers, (or Top-$k$ in particular) for any compressors, the existing literature predominantly focuses on uniform compression throughout the training, where the same compression ratio

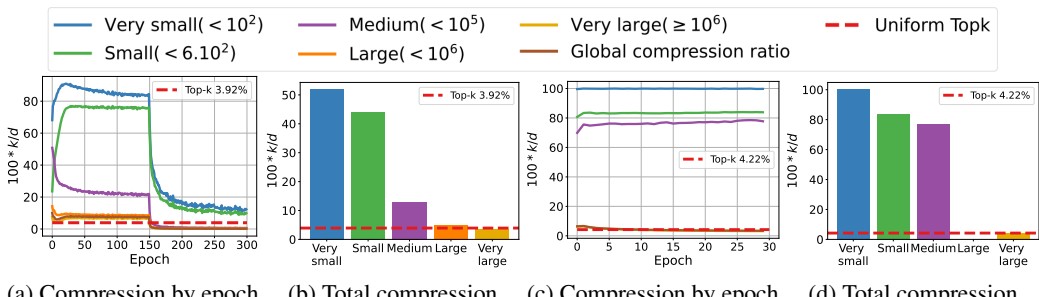

Figure 1: Compression ratio (in percentage, we use $\frac{k}{d} \times 100$) vs. the training iterations and layer size in training ResNet-18 on CIFAR-100 ((a) and (b)) and NCF on MovieLens-20 M ((c) and (d)) using the Top-$k$ and Threshold sparsifiers. For ResNet18 and NCF, $k$ is set to 3.92%, and 4.22%, respectively, and $\lambda = 0.1$.

is used for all layers. Although varying the compression ratio for each layer at different stages of training is feasible, this area is not well-explored and most available literature proposes compute-heavy methods to find the best compressor (Khirirat et al., 2021; Xin et al., 2023; Markov et al., 2024). Attempts were made to achieve optimal compression performance by adopting different adaptive strategies; see §A. In contrast, we investigated Occam's Razor principle: "plurality should not be posited without necessity." Instead of employing computationally intensive strategies, can we provide a simple yet efficient strategy for quickly selecting a compression parameter for each layer, achieving a good balance between compressed data volume and model performance?

In that pursuit, we train two DNN architectures: (*i*) ResNet-18 (He et al., 2016) on CIFAR 100 (Krizhevsky et al., 2009) (baseline no compression Top-1 accuracy is 73.38%) and (*ii*) NCF on MovieLens-20M (Harper & Konstan, 2015) (baseline no compression best Hit-Rate@10 is 95.59%), on standard PyTorch benchmark using 2 NVIDIA A100-SXM4 GPUs with 80 GB memory, connected via 400 Gbps network bandwidth. We use the Top-$k$ and threshold sparsifiers and set the hyperparameters $k$ and $\lambda$ to send the same data volume. While uniform Top-$k$ achieves a Top-1 accuracy of 73.04% on ResNet-18 and a best Hit-Rate@10 of 91.33% on NCF, the threshold sparsifier achieves a Top-1 accuracy of 73.32% on ResNet-18 and a best Hit-Rate@10 of 92.7% on NCF, respectively. To get a better insight into threshold sparsifier's superior performance over the Top-$k$, we plot the compression ratio for different layers of ResNet-18 over iterations and the total average compression of its different size layers; see Figures 1(a)-(b). We observed that the small and medium layers (dimension less than $10^2$ to up to $10^5$) are not so severely compressed during the training compared to the large and very large layers (dimension more than $10^6$)—larger layers experience extremely aggressive compression—even more aggressive than the uniform Top-$k$ for those layers. Additionally, regardless of their sizes, during the beginning phase of the training, the layers are less aggressively compressed compared to the final training phase. We made almost identical observations in the NCF training; see Figures 1 (c)-(d).

Our empirical observations in using the Top-$k$ sparsifier and its adaptive counterpart for DNN training indicate *two key factors*: (*a*) *the layer size of the DNNs* influence in choosing how much one needs to compress, and (*b*) *the training phase of the DNNs* can be a critical contributor in the dynamic compressor design. Moreover, the second observation is consistent with recent research on the *critical training regime of DNNs* (Achille et al., 2019; Agarwal et al., 2021a; Zhang et al., 2022). Although our quest for designing a dynamic compressor primarily started with sparsifiers, the above-mentioned simple factors can be used conjointly with any compressor in designing its efficient counterpart. Based on them, we propose a compression framework that dynamically sets the compression level based on the epoch and layer size. Although both words, adaptive and dynamic, can depict our framework, we prefer to refer to it as dynamic, as we do not use any complicated, hard-to-compute, infeasible gradient statistics during training.

We list our contributions as follows:

**Dynamic compressor scheduler (§2).** We present a Lightweight Efficient GrAdient Compression StrategY or `LEGACY` that, in theory, can work with any compression technique to produce its simple dynamic counterpart. `LEGACY` is based on easy-to-obtain information—layer size and training phase. Designing `LEGACY` is empirically motivated and stands on solid technical intuitions; see §2.1. Irrespective of the DNN models and training dataset, `LEGACY` can guide the selection of compression parameters based on the layer size or training phase; see system design in §C. To simplify hyperparameter selection, we propose a simplified version of `LEGACY` in §C.1 called **Simple**-`LEGACY` or S-`LEGACY`; §C.1.1 demonstrates how the two `LEGACY` approaches can be combined.

**Theoretical insights (§4).** Under the usual assumptions for stochastic first-order algorithms in the compressed, distributed setup, we validate the influence of our policies on the convergence of compressed SGD using biased and unbiased $\delta$-compressors; see Theorem 1 in §4.

**Benchmarking (§5).** We benchmark `LEGACY` through a variety of numerical experiments involving diverse DNN architectures (convolution and residual networks, transformer, recommender system and GPT-2—a total of 7 models) trained for different tasks (image classification on CIFAR 10, CIFAR 100, and ImageNet-1K, text prediction on WikiText-103, OpenWebText, and collaborative filtering on Movielens-20M—a total of 6 datasets; see Table 6 in §D.1) by using Top-$k$, Random-$k$ (sparsifiers), QSGD Alistarh et al. (2017) (quantizer), and PowerSGD (low-rank) Vogels et al. (2019) as base compressors. We report our results using test accuracy, communicated data volume, throughput, and computation time. Additionally, we compared `LEGACY` against 5 state-of-the-art adaptive compressors (CAT Khirirat et al. (2021), Variance-based compression Tsuzuku et al. (2018), Accordion Agarwal et al. (2021a), AdaComp Chen et al. (2018a), and L-Greco Markov et al. (2024)). Finally, in §D.3, we evaluate `LEGACY` in resource-constrained environments (compute and network bandwidth; §D.3.1), federated learning (§D.3.2), and a large-scale configuration with 100 CPU workers (§D.3.3), demonstrating its scalability.

## 1.1 RELATED WORK AND BACKGROUND

Due to limited space, we moved the related work to §A.

**Notations.** We use $\|x\|$ to denote the $\ell_2$-norm of a vector $x$. By $g_{i,t}$ and $\nabla f_{i,t}$, we denote the stochastic gradient and full gradient, respectively, at the $i^{th}$ node at iteration $t$.

**Compressor.** A random operator, $\mathcal{C}(\cdot) : \mathbb{R}^d \to \mathbb{R}^d$ is a *compression operator* if $\mathbb{E}_\mathcal{C}\|x - \mathcal{C}(x)\|^2 \leq (1 - \delta)\|x\|^2$ for all $x \in \mathbb{R}^d$, where $\delta > 0$ is the compression factor. A smaller $\delta$ indicates a more aggressive compression. In our setup, $\delta \in (0, 1]$, and $\mathcal{C}$ is a $\delta$-*compressor*. The popular sparsifiers, $\text{Top}_k$ and $\text{Random}_k$ have $\delta = \frac{k}{d}$, and $\mathbb{E}\|x - \text{Top}_k(x)\|^2 \leq \mathbb{E}\|x - \text{Random}_k(x)\|^2 \leq (1 - \frac{k}{d})\|x\|^2$.

## 2 DESIGNING A DYNAMIC COMPRESSOR

From Figure 1, we observe two key factors in DNN training. First, the compression ratio has more impact at the beginning of training than at the end. Second, it is better to compress large layers and keep small layers uncompressed (or easy compression). But can these observations also be theoretically justified so that we can build a dynamic compressor scheduler based on them?

To answer this, we formulate the impact of unbiased compressors on the decrease rate for the gradient descent (GD) algorithm under two relatively easier-to-analyze cases: (*i*) smooth, strongly convex functions, and (*ii*) smooth, nonconvex functions with PL condition. There is no loss of generality in considering GD instead of distributed SGD — analysis of GD offers ease of notation, and under simple arguments, leads us to a practical scheduler.

**Setup.** Consider the *empirical risk minimization* (ERM) problem with $n$ computing nodes:

$$\min_{x\in\mathbb{R}^d} \left[ F(x) := \frac{1}{n}\sum_{i=1}^{n} f_i(x) \right], \tag{1}$$

where $f_i(x) := \mathbb{E}_{z_i \sim \mathcal{D}_i} l(x; z_i)$ is the loss function at node $i$ on input $z_i$ sampled from its distribution, $\mathcal{D}_i$. Let $g_{i,t}$ be the stochastic gradient computed at the $i^{th}$ node in iteration $t$ and of the form $g_{i,t} = \nabla f_{i,t} + \xi_{i,t}$, with $\mathbb{E}[\xi_{i,t}|x_t] = 0$. We made general assumptions in §B.1 to prove our results.

## 2.1 INSIGHT THROUGH THE LENS OF THE COMPRESSED GD

Let $\mathcal{C}_t$ be unbiased $\delta_t$-compressors for all $t \in [T]$. The iterative update rule of the compressed GD algorithm with fixed stepsize, $\eta \geq 0$ and unbiased $\delta_t$-compressors in solving (1) is given by

$$x_{t+1} = x_t - \eta\mathcal{C}_t(\nabla F(x_t)). \tag{2}$$

In the following lemma, we quantify the decrease in the quantity, $\|x_{t+1} - x_*\|^2$ under the smoothness and strong convexity assumption; see the proof in §B.2.

**Lemma 1.** *Let F follow Assumptions 1 and 2. Then with fixed stepsize $\eta$, the sequence of iterates, $\{x_t\}_{t\geq0}$ of compressed GD updates satisfy*

$$\mathbb{E}_{\mathcal{C}_t}\|x_{t+1} - x_*\|^2 \leq \underbrace{\left(1 - 2\mu\eta + \eta^2\mu L(2 - \delta_t)\right)\|x_t - x_*\|^2}_{D(\delta_t):=Real\ decrease}.$$

**Algorithm 1:** Compressed distributed training without error feedback (EF)

**Input:** Number of nodes $n$, learning rate $\eta$, number of iterations T, batch-size $\mathcal{B}$ per node as $\mathbf{n_{batch}}$
**Output:** The trained model $x$
**for** $t = 0, 1, \ldots,$ T **do**
    **On each node** $i$:
    $g_{i,t} = $ Calculategradient$(x_t, \mathbf{n_{batch}})$
    $k_{i,t} = $ Chooseparam$(g_{i,t}, t)$
    $\tilde{g}_{i,t} = $ Compress$(g_{i,t}, k_{i,t})$
    Communicate$(\tilde{g}_{i,t})$
    **On Master**:
    $[\tilde{g}_{1,t}, \ldots, \tilde{g}_{n,t}] = $ Receive$(\mathbf{n})$
    $[g_{1,t}, \ldots, g_{n,t}] = $
      Decompress$([\tilde{g}_{1,t}, \ldots, \tilde{g}_{n,t}])$
    $g_t = $ AverageGrads$([g_{1,t}, \ldots, g_{n,t}])$
    Broadcast$(\mathbf{g_t})$
    **On each node** $i$:
    $x_{t+1} = $ Update$(x_t, g_t, \eta)$

Table 1: Functions used in our framework.

| Function | Description |
|---|---|
| Chooseparam | Decide compression parameters |
| Compress | Apply compression to each layer |
| Communicate | Send compressed gradient to the server |
| Receive | Gather the compressed gradients from workers |
| Decompress | Restore the original tensor shape |
| AverageGrads | Average the received gradients |
| Broadcast | Broadcast the averaged gradient |
| Update | Optimizer independent parameter update |

Note that $D(\delta_t)$ is a function of the compression factor. For no compression, $\delta_t = 1$, we obtain:

$$\|x_{t+1} - x_*\|^2 \leq \left(1 - 2\mu\eta + \mu\eta^2 L\right) \|x_t - x_*\|^2 := D(1) := \text{Ideal decrease.}$$

Ideally, we are interested in $\delta_t \in (0, 1]$ such that $D(\delta_t)$ (i.e., the compressed GD decrease) is as close as possible to $D(1)$ (i.e., the non-compressed GD decrease). We have

$$\Delta := D(\delta_t) - D(1) = \mu\eta^2 L(1 - \delta_t)\|x_t - x_*\|^2.$$

To have $\Delta \approx 0$, we require:

(*i*) **Strategy I: Compression based on the training phase.** At the beginning of the training, we have $\|x_t - x_*\|^2 \gg 0$. Therefore, to make $\Delta \approx 0$ we need to choose $\delta_t \to 1$ (no or easy compression). At the end of the training, $\|x_t - x_*\|^2 \approx 0$. Hence, no strong control is needed on $\delta_t$ to keep $\Delta$ small. In this case, one can choose $\delta_t \approx 0$ (aggressive compression).

(*ii*) **Strategy II: Compression based on the layer sizes.** We observed that a small subset of layers in DNN training dominates the communication overhead. E.g., for the models benchmarked in this paper, the largest 20% of layers account for $\sim 90\%$ of the total parameter volume, while the remaining 80% contribute to only 10%. This makes large layers ideal candidates for aggressive compression, as they are usually overparameterized with more redundancy and can tolerate higher compression without significantly affecting model quality.

In contrast, smaller layers, while contributing little to total communication volume, often contain gradients that are critical for stable and efficient convergence. Maintaining higher fidelity for these gradients is therefore important. By compressing large layers more aggressively and reallocating the resulting bandwidth savings to smaller, more sensitive layers, we can reduce compression severity where accuracy is most impactful, thereby improving overall gradient quality without increasing total communication cost. E.g., for Top-$k$, the contribution of compressed elements from large layers to the quantity $\|x_t - x_\star\|$ is small. The small layers with less redundancy than the large layers are crucial, and their *suboptimality* (if compressed too much) may lead to larger errors. Therefore, compared to uniform $\delta$-compressors, with strategy II, it is better to easily or not compress small layers, i.e., $\delta_s \approx 1$, and focus on compressing large layers with compression $\delta_l \approx \delta$. With this strategy, we result in similar data volume and improved convergence. Even a slight increase in compression aggressiveness for large layers, such as reducing the transmitted gradient volume from Top-$k$ with $k = 10\%$ to $k = 9.95\%$, can yield a substantial benefit. While the 0.05% reduction is negligible for a large layer in terms of both volume and performance impact, the reclaimed budget can represent a significant increase for smaller layers, potentially allowing a shift from transmitting just 10% to over 50% of their gradients; in some cases, it can be a 100% reduction or no compression. Such redistributions dramatically enhance gradient representation in small layers, improving convergence stability and overall model accuracy with minimal trade-offs; see a practical example and another angle to look at Strategy II in §C.1.3.

To further extend our theoretical insight for Strategy-I, in the next lemma, we consider GD for minimizing a smooth nonconvex function under the PL condition and quantify the functional suboptimality gap, $E_{\mathcal{C}_t}(F_{t+1}) - F_*$; see the proof in §B.2.

**Function 2:** `EpochCompression(`$\{\lambda_i\}_{i=1}^p, \{\delta_i\}_{i=1}^p$`)`

**Input:** Current iteration, $t$
**Output:** Compression parameter, $\delta_i$
$j = $ `index` of the smallest threshold from $\{\lambda_i\}_{i=1}^p$ such that iteration $t \leq \lambda_i$ ;
**return** $\delta_j$

**Function 3:** `LayerSizeCompression(`$\{\lambda_i\}_{i=1}^p, \{\delta_i\}_{i=1}^p$`)`

**Input:** Gradient $g_{i,t}$ at iteration $t$ from worker $i$
**Output:** `compression parameters list`
**for** *each layer $L$ in $g_{i,t}$* **do**
  $\quad j = $ `index` of the smallest threshold from $\{\lambda_i\}_{i=1}^p$
  $\quad\quad$ such that $|L| \leq \lambda_j$;
  $\quad$ Append $\delta_j$ to `compression parameters list`;
**return** `compression parameters list`;

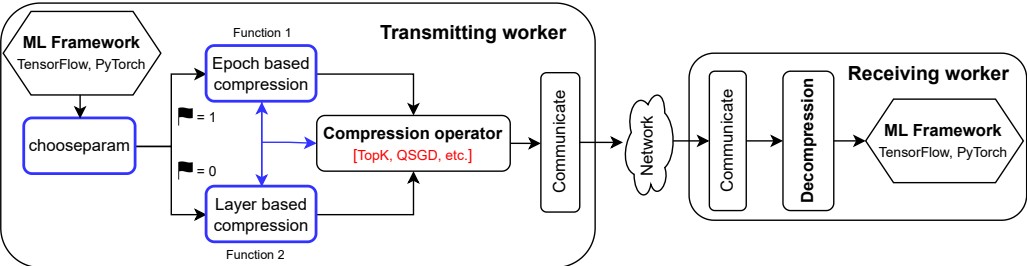

Figure 2: System architecture. The `LEGACY` framework is highlighted in blue.

**Lemma 2.** *Let $F$ follow Assumptions 1 and 4. Then with stepsize $\eta = \frac{1}{L}$, the sequence of iterates, $\{x_t\}_{t\geq 0}$ of compressed GD updates satisfy $E_{\mathcal{C}_t}(F_{t+1}) - F_* \leq \left(1 - \frac{\delta_t \mu}{L}\right)(F_t - F_*) := D(\delta_t) := $ Real decrease.*

Substituting $\delta_t = 1$ gives the ideal decrease, i.e., the decrease in the functional suboptimality gap without compression: $F_{t+1} - F_* \leq \left(1 - \frac{\mu}{L}\right)(F_t - F_*) := D(1) := $ Ideal decrease. To have $\Delta := D(\delta_t) - D(1) = (1 - \delta_t)\frac{\mu}{L}(F_t - F_*) \approx 0$, we require: (*i*) At the beginning of the training $F_t - F_* \gg 0$. Therefore, we need to choose $\delta_t \approx 1$ (no or easy compression) to keep $\Delta \approx 0$. (*ii*) At the end of the training $F_t - F_* \approx 0$. Therefore, we can choose $\delta_t \approx 0$ (aggressive compression).

## 2.2 A DYNAMIC COMPRESSOR SCHEDULER

Motivated by the previous section, we formally define a dynamic compressor scheduler for compressed distributed training on $n$ workers. Although our scheduler is optimizer agnostic, for simplicity, we consider the optimizer to be SGD. Given a stepsize sequence, $\{\eta_t \geq 0\}_{t\geq 0}$ and $\delta_t$-compressors, the update rule for compressed distributed SGD on $n$ workers is given by

$$x_{t+1} = x_t - \frac{\eta_t}{n} \sum_{i=1}^n \mathcal{C}_t(g_{i,t}). \tag{3}$$

Algorithm 1 provides a general compressed communication framework without error feedback (Karimireddy et al., 2019). Our approaches build on this framework by adjusting the compression level via the `chooseparam` function; unimodal training in blue. We require two user-inferred hyperparameters: (*i*) a sorted list of $p$ decreasing compression levels, $\{\delta_i\}_{i=1}^p$, of the $\delta$-compressor $\mathcal{C}_t$, where $\delta_p$ being the most aggressive compression factor, and (*ii*) a sorted list of $p$ non-decreasing thresholds, $\{\lambda_i \geq 0\}_{i=1}^p$, which represents either an iteration or a layer size at which we use a certain compression level $\delta_i$, in Algorithm 1. The threshold change is based on the following approaches:

(*i*) **Training epoch dependent.** We start with a less intense compression and gradually increase its intensity during the training. In `Epoch compression`, we progressively increase the compression level $\delta$ as training progresses; see Function 2. In this case, the non-decreasing thresholds $\{\lambda_i\}_{i=1}^p$ denote the iterations or epochs at which the compression intensity increases.

(*ii*) **Layer size.** We employ an easy compression for small layers as their size is insignificant compared to the larger ones. We achieve this through `LayerSizeCompression`; see Function 3. In this Function, we used the thresholds $\{\lambda_i\}_{i=1}^p$ to group layers by their sizes; smaller layers are affected by a less intense compression, while the larger layers experience a more aggressive compression.

## 3 SYSTEM ARCHITECTURE, HYPERPARAMETER CHOICE, LEGACY VARIANTS

We present Lightweight Efficient GrAdient Compression StrategY or `LEGACY`; illustrated in Figure 2. `LEGACY` is framework-agnostic (e.g., compatible with TensorFlow and PyTorch) and provides

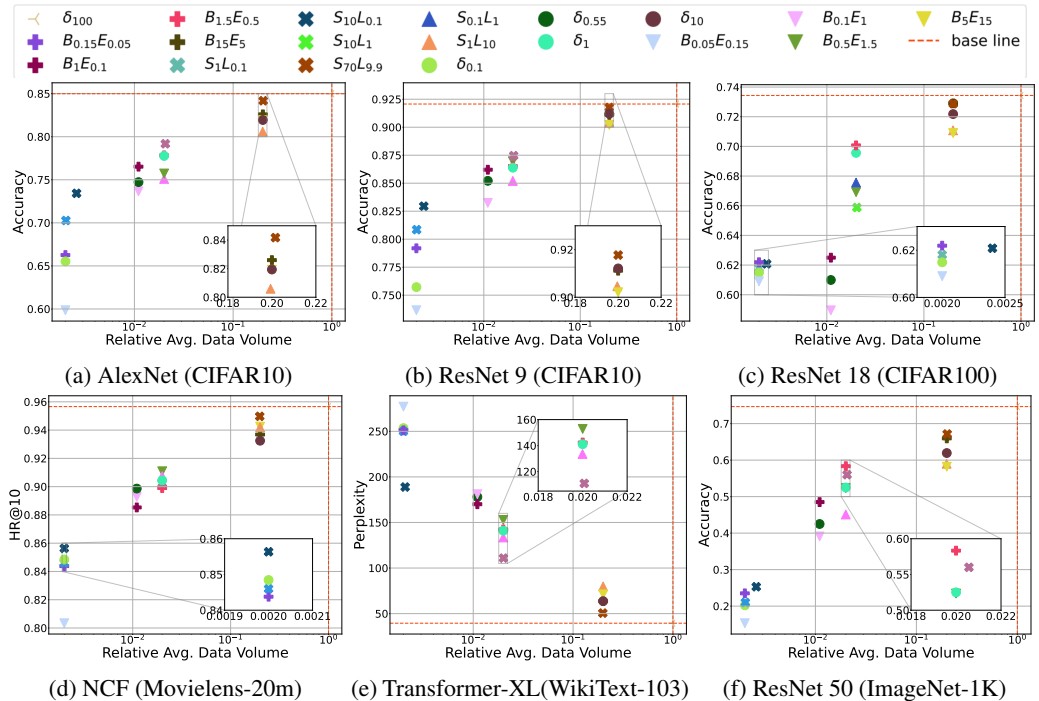

Figure 3: Layer-size and training epoch dependent Top-$k$ and uniform Top-$k$ (denoted by $\delta_{\text{compression}}$) — Relative average data volume vs. model quality.

a simple API that supports arbitrary gradient compressors such as Top-$k$ and QSGD. For clarity, Algorithm 1 instantiates `LEGACY` within a parameter-server architecture; however, this serves only as a proof of concept. `LEGACY` is independent of both the underlying base compression method and the communication protocol. We conducted Top-$k$ with NCCL `AllGather` NCCL, demonstrate scalability via CPU-based experiments using `Gloo` gloo in §D.3, and use `AllReduce` for PowerSGD experiments in §5.4. Since `LEGACY` only governs compression parameter selection, it integrates seamlessly into any distributed training setup (e.g., parameter-server, ring all-reduce) where gradient compression is applicable.

For transmitting workers, `LEGACY` is executed through the intermediary API call `chooseparam` in Algorithm 1, responsible for selecting the appropriate compression parameters for each layer. After gradient computation through any ML benchmark, based on the user's strategy, epoch compression Function 2 (🏳 = 1) or Layer size compression Function 3 (🏳 = 0) is invoked to dynamically determine the compression parameters for each layer, which are then applied to the gradient compressor in the worker. Additionally, Functions 2 and 3 in `LEGACY` can be used conjointly with the base-compressor; see the blue three-point arrow. Other than `chooseparam`, `LEGACY` uses other well-known APIs for communication, averaging, broadcasting, etc.; see Table 1. The receiving worker applies reverse operations and decompresses the received gradient. `LEGACY` can be used for uplink and downlink bidirectional compression by simply compressing the gradient sent from the server.

To simplify the selection of hyperparameters used in LEGACY ($\{\lambda_i\}_{i=1}^p$ for thresholds and $\{\delta_i\}_{i=1}^p$ for compression levels), we propose a simplified version of `LEGACY` in §C.1 called **Simple**-`LEGACY` or S-`LEGACY`. This version requires only two hyperparameters for the epoch or layer-based approach and three for the mixed approach. That is, S-`LEGACY`-E requires only a default compression parameter $\delta_u$ and the number of training phases $n$; S-`LEGACY`-L requires $\delta_u$ and a decrease ratio $s$; S-`LEGACY`-M (stands for using both layer and epoch-based) in §C.1.1 demonstrates how the two LEGACY approaches can be combined and it requires $\delta_u$, $n$, and $s$.

## 4 CONVERGENCE GUARANTEE

In this Section, we establish the nonconvex convergence of distributed SGD with $\delta_t$-compressors. To prove a general distributed convergence with a $\delta$ compressor, biased or unbiased, we want to estimate the quantity, $\mathbb{E}\left[\left\|\frac{1}{n}\left(\sum_{i=1}^n \mathcal{C}_t(g_{i,t})\right)\right\|^2 | x_t\right]$, as this is highly influenced by `LEGACY` 's strategies. To achieve that, we adopt slightly different approaches for biased and unbiased compressors, $\mathcal{C}_t$.

Denote $\beta_t := (1 - \delta_t)(M + 1) + M$, where $M, \sigma^2 \geq 0$ are constants such that for all $x_t \in \mathbb{R}^d$, the stochastic noise, $\xi_{i,t}$ follows $\mathbb{E}[\|\xi_{i,t}\|^2 \mid x_t] \leq M\|\nabla f_{i,t}\|^2 + \sigma^2$; see Assumption 5. The constants appearing are due to the general Assumptions in §B.1. For biased compressors, we adopt a few extra assumptions. First, we consider a $(C, \zeta^2)$ bounded similarity assumption on the variance of the gradients among the workers. This Assumption is stronger than Assumption 6, but a similar assumption was proposed in Sahu et al. (2021), and we rely on it for algebraic purposes in proving the convergence for the general case. Due to limited space, we defer the biased-compressor results, including intermediate Lemmas and the main complexity theorem, to §B.3.2.

**Lemma 3.** *(Compression Bounds) Let the stochastic noise follow Assumption 5. Let $\mathcal{C}_t$ be unbiased $\delta_t$-compressors for all $t \in [T]$, and let $F$ follow Assumption 6. We have $\mathbb{E}\left\|\frac{1}{n}\sum_{i=1}^{n}\mathcal{C}_t(g_{i,t})\right\|^2 \leq \frac{2A\beta_t}{n}(F_t - F_\star) + \left(1 + \frac{\beta_t}{n}\right)\|\nabla F_t\|^2 + \frac{B\beta_t}{n} + \left(\frac{2 - \delta_t}{n}\right)\sigma^2$.*

Using the previous Lemma, the following theorem gives the complexity results for unbiased $\delta_t$ compressors, which are similar to the classical complexity results for compressed SGD-type algorithms; see Dutta et al. (2020); Stich & Karimireddy (2020); Sahu et al. (2021). The proof is given in §B.3.

**Theorem 1.** *(Nonconvex convergence) (*i*) Let Assumptions 1, 3, 5, and 6 hold. Let $\mathcal{C}_t$ be unbiased $\delta_t$-compressors for all $t \in [T]$. For a stepsize $\eta \leq \min\left(\frac{1}{\frac{L}{2} + \frac{L(2M+1)}{n}}, \left(\frac{AL(2M+1)T}{n}\right)^{-\frac{1}{2}}\right)$ we have:*

$$\min_{t=0,1,\cdots T-1}\mathbb{E}\|\nabla F_t\|^2 \leq \frac{3(F_0 - F_\star)}{T\eta\left(1 - \frac{L\eta}{2} - \frac{L\eta(2M+1)}{n}\right)} + \hat{\sigma}, \text{ where } \hat{\sigma} = \frac{L\eta\left(B(2M+1) + 2\sigma^2\right)}{2n\left(1 - \frac{L\eta}{2} - \frac{L\eta(2M+1)}{n}\right)}.$$

Theorem 1 primarily guarantees that the (nonconvex) convergence of distributed SGD with $\delta_t$-compressors in each iteration (which the policies of LEGACY govern: a changing compression level over iteration and $\delta_t$ modulates the effect of epoch- or layer-based strategies). The factor, $\hat{\sigma} = \frac{L\eta\left(B(2M+1) + 2\sigma^2\right)}{2n\left(1 - \frac{L\eta}{2} - \frac{L\eta(2M+1)}{n}\right)}$, is the so-called *variance*, incurred due to the effect of multiple factors, including compression ratio, $\delta_t$. However, the theoretical analysis does not distinguish the relative performance of the epoch- or layer-based strategies.

## 5 BENCHMARKING AND EVALUTAION

**Environment and Configuration.** We run our experiments on 4 NVIDIA A100-SXM4 GPUs (2 GPUs for AlexNet, ResNet-9, ResNet-18, and GPT-2 training, and 4 GPUs for Transformer-XL, NCF, and ResNet-50 training) with 80GB memory and interconnected with 400 GBps bandwidth. LEGACY is built on Dutta et al. (2020); Sahu et al. (2021); for Transformer-XL, we used the NVIDIA Training Examples benchmark Nvidia with reduced steps; CIFAR10, CIFAR100 and NCF tests were implemented using Dutta et al. (2020), Sahu et al. (2021), and Nvidia, respectively. We used 30 epochs for AlexNet, ResNet-9, and NCF training, 300 epochs for ResNet18 training, and 4,500 steps for the Transformer training. For ImageNet-1K, we employed PyTorch and trained ResNet-50 for 50 epochs. We follow karpathy Andrej (2023) implementation for our GPT-2 experiments and use 100K training iterations; see Tables 6 and 7 in §D.1 for a detailed summary and Tables 9–14 in §D.1 reproducibility.

**LEGACY Setup.** In the main paper, for simplicity, we split the training into two phases: beginning $B$ (first half of the total epochs) and end $E$ (rest of the total epochs); each phase uses a different compression level. For layer sizes, we categorize layers into two groups: small layers, $S$ with fewer than $10^4$ elements, and large layers, $L$ with $10^4$ elements or more. With this formalization, $S_{\delta_1}L_{\delta_2}$ means small layers are compressed with compression factor, $\delta_1$ and large layers compressed with compression factor, $\delta_2$, and $B_{\delta_1}E_{\delta_2}$ denotes two-phase training, beginning phase with compression factor, $\delta_1$, and end phase with $\delta_2$. In §D.2, we show the efficacy of LEGACY by adding multiple training phases and more layer granularity.

### 5.1 MODEL QUALITY VS. TRANSMITTED DATA VOLUME

Figure 3a shows the accuracy of AlexNet on CIFAR-10; uniform Top-$k$ compression with $k = 0.1\%d$ (corresponding to the $\delta_{0.1}$) results in an accuracy of 75.7%. However, using Top-$k$ as base compression in LEGACY, the strategy, $B_{0.15}E_{0.05}$, which starts with a compression ratio of 0.15% for the first half of the epochs and then switches to an aggressive compression ratio of 0.05%, achieves a higher accuracy of 79.18%. Notably, the reverse strategy $B_{0.05}E_{0.15}$ results in a lower accuracy of 73.6%. When we compress smaller layers at 1% while keeping the larger layers at the 0.1% ratio, $S_1L_{0.1}$, the accuracy improves by 5.14% over the uniform compression. Figures 3b – 3f show similar results across different DNN models and challenging, larger datasets, including

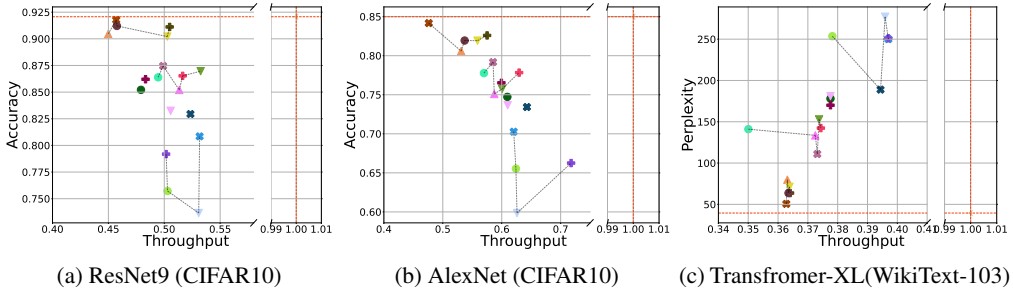

(a) ResNet9 (CIFAR10)  (b) AlexNet (CIFAR10)  (c) Transfromer-XL(WikiText-103)

Figure 4: Layer-size and training epoch dependent Top-$k$ and uniform Top-$k$ — Throughput vs. model quality, where experiments with similar global compression ratios are linked with a dotted line; see legend in Figure 3.

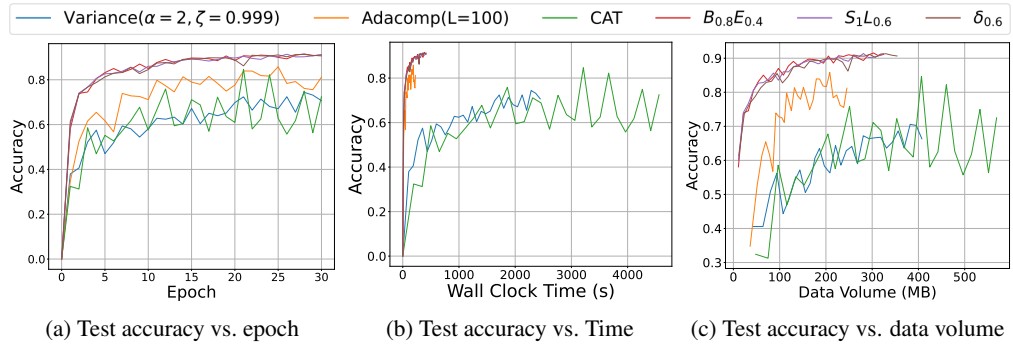

(a) Test accuracy vs. epoch  (b) Test accuracy vs. Time  (c) Test accuracy vs. data volume

Figure 5: Comparison of LEGACY with Top-$k$ and other adaptive compressors in training ResNet9 on CIFAR10.

ImageNet-1K and WikiText, with accuracy improvements up to 7-11% on ImageNet-1K compared to the uniform compression strategy. For language model in Figure 3e, the perplexity improves $\sim 26\%$, from 253.57 with uniform $\delta_{0.1}$ to 188.8 with dynamic compression $S_{10}L_{0.1}$.

**Takeaways.** For comparable data volumes, starting with mild compression and gradually increasing it outperforms uniform or inverse strategies by allowing DNN models to retain crucial information during early training phases. This approach balances the need for sufficient data in the early stages with the efficiency of higher compression later. Similarly, leaving small layers uncompressed or lightly compressed results in only a minor increase in data volume but improves perplexity by 26% on WikiText-103 and accuracy by 7% on ImageNet-1K.

## 5.2 MODEL QUALITY VS. TRAINING THROUGHPUT

Figure 4 shows the impact of compression on model quality as a function of the relative throughput. Test cases with a similar average compression ratio ($\pm 10\%$) are connected with dotted lines. The throughput under compression is lower than the no-compression baseline because the workers are connected through high-bandwidth links, making the compression overhead relatively higher than the communication cost. Analyzing the groups (connected by the dotted lines), we observe that the average compression ratio influences the model performance and throughput; sending more data improves accuracy but reduces throughput. Applying moderate compression during the initial training phase and to smaller layers yields better performance for a similar average compression ratio. In Figure 4a, for ResNet9, a uniform Top-0.1% compression results in 75% accuracy, and 50.29% relative throughput. while our epoch-based strategy, $B_{0.15}E_{0.05}$, yields similar relative throughput but improved accuracy, reaching 79.18%. Meanwhile, the layer size-based strategy, $S_1L_{0.1}$, further improves throughput to 53.16% and accuracy to 80.85%, yielding gains of 5.7% in throughput and 6.6% in accuracy compared to the uniform compression. We observe similar findings in Figures 4b and 4c. Generally, the dynamic strategies in LEGACY (denoted by '+' for epoch-based and '×' for layer size-based) for linked points are positioned either above (indicating better accuracy) or to the right (indicating better throughput) of the uniform case for AlexNet and ResNet9. For the Transformer-XL, LEGACY strategy points are located to the right of or below the uniform case, under similar average compression ratios, indicating a better perplexity, with improvements of up to $\sim 26\%$ in perplexity and $\sim 4.5\%$ in throughput compared to uniform compression.

**Takeaways.** Our layer-based strategy can increase accuracy and throughput compared to the uniform or inverse approaches, although the throughput gains are limited due to the high-speed network in the data center. For the layer size-based approach, not compressing small layers eliminates the

Table 2: Comparison of `LEGACY` with uniform QSGD on training ResNet9 on CIFAR 10.

| Methods | Uniform QSGD Alistarh et al. (2017) | LEGACY-E | LEGACY-L |
|---|---|---|---|
| Top-1 Accuracy | 87.21 | 87.98 | **88.42** |
| Avg. relative data volume | 18.78% | 18.76% | **16.4%** |

computational overhead. For the epoch-based approach, sending more data at the beginning appears to balance out the aggressive communication towards the end, yielding similar throughput while leveraging the early training stages to achieve better accuracy.

### 5.3 COMPARISON WITH ADAPTIVE GRADIENT COMPRESSORS

We evaluate our approaches, $B_{\delta_1}E_{\delta_2}$ and $S_{\delta_1}L_{\delta_2}$, using Top-$k$ in `LEGACY` against three state-of-the-art adaptive compressors (Adacomp Chen et al. (2018a), variance-based compression Tsuzuku et al. (2018), and CAT Khrirat et al. (2021)) in terms of the trained model quality and the training time. As shown in Figures 5a–5b, our scheduler achieves higher accuracy at comparable data volumes. Although we are slower than AdaComp, which is threshold-based and $2\times$ faster than the uniform Top-$k$ and our strategies, we achieve a 12% accuracy gain while sending only $\sim$75MB more data; see Figure 5c. Variance-based compression requires access to per-sample gradients, which are not supported by most deep learning frameworks; obtaining these values using a batch size of one is extremely slow. We used OPACUS Yousefpour et al. (2021) to get faster per-sample gradients. Still, it remains $\sim 6\times$ slower than our approaches with a 15% lower accuracy. CAT requires testing many values at each iteration before choosing the sparsity, resulting in $11\times$ slower performance, sending around 575Mb of data, and incurring 25% lower accuracy than our approaches. Our strategies are robust as they choose the compression ratios and control the total and per-iteration data volume. In contrast, except for Accordion, other adaptive methods can neither be applied to different compressors nor provide an estimate of the data volume. We also found that at the core, these methods exhibit similar behavior to our strategies, confirming the effectiveness of our approach, which does not require additional computation. See the complexity results in D.4.

### 5.4 QUANTIZATION AND LOW-RANK FACTORIZATION WITH LEGACY

**QSGD Alistarh et al. (2017) experiment details and results.** We train ResNet-9 on CIFAR-10 using 2 workers for 30 epochs. QSGD has one *user-defined* parameter $s \geq 1$, the quantization level. For uniform QSGD, we fix $s = 32$. For `LEGACY` we introduce more granularity, we use 3 groups to represent three different training phases, the beginning of training (1-10 epoch, with $s = 64$), the middle of training (11-20 epoch, with $s = 32$), and the end of training (21-30 epoch, with $s = 16$); and 4 groups to represent four distinct layer sizes, very small layer, $S(< 600)$ are left uncompressed, medium-sized layers, $M (< 100,000$ with $s = 256)$, large layers, $L (< 1,000,000,$ with $s = 64)$ and huge layers, $H (\geq 1,000,000,$ with $s = 16)$. Table 2 shows `LEGACY-L` renders a 1.39% accuracy gain relative to uniform QSGD by sending about 12.67% less data.

**PowerSGD Vogels et al. (2019) experiment details and results.** We train ResNet-18 on the CIFAR-100 dataset using 2 workers for 200 epochs with PowerSGD as the base compressor, and we compare `LEGACY` with two adaptive compressors, Accordion Agarwal et al. (2021a) and L-GreCo Markov et al. (2024). PowerSGD has one user-defined parameter, rank $(r)$. The smaller the rank, the more aggressive the compression is. For uniform PowerSGD, we use $r = 3$, for L-GreCo and Accordion, we kept the same configuration as in their public implementation. We note that in those implementations, to improve accuracy, the authors used no compression for the first 1,000 iterations. This strategy leads to a higher volume of transmitted data. For `LEGACY-E`, we use 4 groups to represent 4 different training phases, the first quartile of training Q1 (1-50 epoch, with $r = 6$), the second quartile Q2 (51-100 epoch, with $r = 4$), third quartile Q3 (101-150 epoch, with $r = 3$), and the final quartile Q4 (150-200 epoch, with $r = 2$); and for `LEGACY-L` (stands for layer-based) we use 4 groups to represent 4 distinct layer sizes, small layer, $S(< 600)$ are left uncompressed, medium-sized layers, $M (< 100,000$ with $r = 8)$, large layers, $L (< 1,000,000,$ with $r = 3)$ and huge layers, $H (\geq 1,000,000,$ with $r = 2)$. Table 3 shows the results on PowerSGD combined with `LEGACY`. Uniform rank-3 PowerSGD transmits about 2.85% of the total data volume and achieves 74.58% test accuracy. All adaptive compressors outperform uniform rank-3 PowerSGD, albeit by a smaller margin. Interestingly, both strategies in `LEGACY` outperform the adaptive compressors while only sending half of their communicated data volume; see Table 3. Moreover, in contrast to L-GreCo, `LEGACY` is compute-free; it does not need to calculate optimal parameters per layer at each call. Our results demonstrate that `LEGACY` is compatible with different compression paradigms, and show its flexibility and versatility in handling different compression techniques.

Table 3: Comparison of `LEGACY` and adaptive compressors on ResNet-18 training on CIFAR-100 using PowerSGD as the base compressor. The ↓ arrows indicate the relative data-volume gain compared to the baseline PowerSGD, while the ↑ arrows indicate the performance gain compared to the baseline PowerSGD. By sending way less data compared to PowerSGD and other SOTA adaptive compressors, L-GreCO and Accordion, `LEGACY` achieves a superior performance.

| Metric | PowerSGD | L-GreCo | Accordion | LEGACY-E | LEGACY-L |
|---|---|---|---|---|---|
| Top-1 Acc. | 74.58 | 75.23 (↑0.87%) | 75.11 (↑0.71%) | **75.55** (↑1.3%) | 75.21 (↑0.84%) |
| Avg. data volume | 2.85% | 2.19% (↓23.16%) | 2.02% (↓29%) | 1.11%(↓61%) | **1.1%**(↓61%) |

Table 4: Training GPT-2 with `LEGACY` scheduler; validation loss the lower the better. The ↓ arrows indicate the relative improvement compared to the baseline uniform Top-$k$.

| Metric | Uniform Top-k | LEGACY-E | LEGACY-L |
|---|---|---|---|
| Validation Loss | 3.14 | 3.05 (↓2.87%) | **3.01**(↓4.14%) |
| Avg. relative data volume | 5% | 5% | 5.02% |

## 5.5 EVALUATING `LEGACY` ON GPT-2

In addition to evaluating `LEGACY` on classical benchmarks, which include ResNet-9, ResNet-50, and Transformer-XL models trained on CIFAR-100, ImageNet-1K, and WikiText-103, we further evaluate `LEGACY` in a modern setting by training GPT-2 small (124M parameters) Radford et al. (2019) on the OpenWebText corpus ($\approx 40GB$) Gokaslan & Cohen (2019) using 2 A100 GPUs. For uniform Top-$k$, we fix the sparsification ratio at 5%. For `LEGACY` layer-based, we introduce structural granularity by grouping layers by parameter count: small layers ($< 1,000$) are left uncompressed, medium layers ($< 1,000,000$) transmit the top 10%, large layers ($< 10,000,000$) transmit 5%, and the largest layers ($\geq 10,000,000$) transmit 4% of their updates. For `LEGACY` epoch-based, the sparsification ratio varies across training phases: 10% for the first 10% of iterations, 7% for the next 10%, followed by 5% and 4% across the next two 30% segments, and 3% for the remainder. Table 4 summarizes the resulting validation losses. Both `LEGACY` variants outperform the uniform Top-$k$ while transmitting a similar amount of data, with the layer-based strategy yielding the best results.

## 5.6 ADDITIONAL BENCHMARKING AND DISCUSSIONS

Due to limited space, we perform a diverse set of experiments in §D that demonstrate the efficacy, scalability, and ease of use of `LEGACY` in different scenarios. This Section serves as a guide to them. We use Random-$k$ as the base compressor in `LEGACY` and show accuracy vs. data volume results in §D.2.1, Figure 6. §D Table 8 reports the average Top-1 test accuracy of ResNet9 and AlexNet on CIFAR10, derived from 7 independent runs; the results are in agreement with §5.1. By using Top-$k$ as the base compressor (with and without error feedback) in `LEGACY`, we provide the model quality vs. wall clock time results in §D.3.1.The model accuracy depends on the compressed gradients being transmitted, not on their transfer speed; the results are generalizable across different bandwidths. In §D.3.3, we perform the scalability of `LEGACY` to 100 workers in a data center environment without constraining the network bandwidth. We also note that fast networks do not always harvest the compression benefit (Xu et al., 2021a), as compression overhead can be significant; in §D.3, we show the efficacy of `LEGACY` in bandwidth-limited federated training. See the limitations and future direction in §E.

## 6 CONCLUSION

We introduce `LEGACY`, an open-source, lightweight framework for dynamic gradient compression in distributed DNN training. In contrast to the compute-intensive adaptive compressors, `LEGACY` operates *dynamically* based on two simple factors—layer size and training phase—and provides a simple yet efficient dynamic scheduler for any compressors. Our benchmarking of `LEGACY` using Top-$k$, Random-$k$, QSGD, and PowerSGD as base compressors show consistent performance gains compared to the uniform compressors and five other state-of-the-art adaptive compressors across large and challenging datasets, including ImageNet-1K and OpenWebText. Finally, in bandwidth-constrained federated training, we profile the efficacy and scalability of `LEGACY` and establish the need of a simple, dynamic scheduler.

**Acknowledgment.** Aritra Dutta is partially supported by the Florida Department of Health Grant, AWD00007072, and the National Science Foundation Grant, 2321986. The authors gratefully acknowledge the support and computing resources from the Toubkal Supercomputer Kissami et al. (2025) at UM6P, Morocco.

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

CONTENTS

APPENDIX

**Organization.** We organized the Appendix as follows: In Section A, we detail the related work. In Section B, we quote the general assumptions and provide detailed proofs of our theoretical results quoted in the main paper. We present the system architecture — `LEGACY`, hyperparameter selection, different variants of `LEGACY`, including simplified LEGACY in Section C. Section D discusses additional numerical results. Finally, in Section E, we discuss the limitations and future research directions.

## A  RELATED WORK

**Gradient compression techniques** are broadly divided into four classes: quantization (Dettmers, 2015; Alistarh et al., 2017; Wen et al., 2017; Bernstein et al., 2018), sparsification (Aji & Heafield, 2017; Stich et al., 2018; Alistarh et al., 2018), low-rank (Wang et al., 2018; Yu et al., 2018; Vogels et al., 2019), and hybrid (Strom, 2015; Dryden et al., 2016; Basu et al., 2019).

**Adaptive compression.** L-Greco Markov et al. (2024) utilizes dynamic programming to determine the optimal compression parameter for each layer under a fixed communication budget. Kimad Xin et al. (2023) and ACE Wang et al. (2024) dynamically monitors network bandwidth instead of using a fixed communication budget; CAT Khirirat et al. (2021) employs a communication cost model to optimize compression efficiency per communicated bit at each iteration.Achille et al. (2019) emphasizes model sensitivity in a certain period, Accordion Agarwal et al. (2021a) aims to identify and respond to this regime by applying a lighter compression during the critical periods. Conversely, LAGS-SGD Shi et al. (2020), and COVAP Meng et al. (2023) take a different approach by adjusting the compression level to overlap gradient communications with computational tasks.

Among less compute-intensive strategies, Luo et al. (2021) decides the compression based on a probability that depends on the gradient value and the layer size. SDAGC Chen et al. (2020b) adjusts thresholds based on the standard deviation of gradients of each layer. AdaComp Chen et al. (2018a) is similar to the threshold compressor, divides gradient components into bins and selects significant components relative to the maximum value in each bin. Guo et al. (2020) determines the quantization level based on the gradient's mean-to-standard deviation ratio; DAGC Lu et al. (2023) assigns compression ratios to workers based on the data distribution. DLS Zhang et al. (2023a) tries to find a layer-wise Top-$k$ compression level. AdapTop-$k$ Ruan et al. (2023) sends more components at the beginning and end of the training and fewer components in the middle. Chen et al. (2018b); Wang et al. (2022; 2023); Deng et al. (2024) suggest freezing or skipping some layers based on their deviation from the previous iteration or by evaluating the importance of the learning of each layer. It can reduce communication and computation by avoiding the gradient computation for the first layers Miyauchi et al. (2018); Wang et al. (2022). Chen et al. (2020a); Qu et al. (2024) compress up and downlink communication.

**Transition to low-bandwidth network.** Compute-intensive techniques such as CAT Khirirat et al. (2021) face performance trade-offs, particularly in fast network environments Agarwal et al. (2021b). In such cases, using basic compressors might take longer than no compression baselines Eghlidi & Jaggi (2020); Xu et al. (2021a); Zhang et al. (2023b). The scenario changes in federated learning (FL) (Kairouz et al., 2021; Xu et al., 2021b; Bergou et al., 2023; Sun et al., 2024), where a low-bandwidth heterogeneous network is de facto. Hence, compression becomes necessary, but employing complex adaptive compressors may reduce the data-saving advantages in FL, especially

when weaker nodes are involved. As a result, we need to focus more on lightweight and simple principles to achieve adaptive compression.

## B  THEORETICAL RESULTS

This section complements Sections 2 and 4 in the main paper. We start with the Assumptions used in the main paper.

### B.1  ASSUMPTIONS

We make the following general assumptions.

**Assumption 1.** *(**Smoothness**) The loss function $f_i : \mathbb{R}^d \to \mathbb{R}$ at each node $i \in [n]$ is L-smooth, i.e. $f_i(y) \leq f_i(x) + \langle \nabla f_i(x), y - x \rangle + \frac{L}{2}\|y - x\|^2$ for all $x, y \in \mathbb{R}^d$.*

**Assumption 2.** *(**$\mu$-strongly convex**) The loss function $f_i : \mathbb{R}^d \to \mathbb{R}$ at each node $i \in [n]$ is $\mu$-strongly convex, i.e. $f_i(y) \geq f_i(x) + \langle \nabla f_i(x), y - x \rangle + \frac{\mu}{2}\|y - x\|^2$ for all $x, y \in \mathbb{R}^d$.*

**Remark 1.** *The above two assumptions together imply that $F$ is L-smooth and $\mu$-strongly convex.*

**Assumption 3.** *(**Global minimum**) There exists $x_\star$ such that, $F(x_\star) = F_\star \leq F(x)$, for all $x \in \mathbb{R}^d$.*

**Assumption 4.** *(**Polyak-Lojasiewicz Condition**) The function $F$ satisfies Polyak-Lojasiewicz (PL) condition with parameter $\mu \geq 0$ if for all $x \in \mathbb{R}^d$ the following holds:*

$$\frac{1}{2}\|\nabla F(x)\| \geq \mu(F(x) - F_*).$$

**Assumption 5.** *(**$(M, \sigma^2)$ bounded noise**) There exist constants $M, \sigma^2 \geq 0$, such that for all $x_t \in \mathbb{R}^d$, the stochastic noise, $\xi_{i,t}$ follows*

$$\mathbb{E}[\|\xi_{i,t}\|^2 \mid x_t] \leq M\|\nabla f_{i,t}\|^2 + \sigma^2.$$

**Remark 2.** *The above implies, $\mathbb{E}[\|g_{i,t}\|^2 \mid x_t] \leq (M+1)\|\nabla f_{i,t}\|^2 + \sigma^2$.*

**Assumption 6.** *(**Bounded variance of gradients**) There exist constants $A, B \geq 0$ such that, for all $x \in \mathbb{R}^d$, the variance of gradients among nodes follow*

$$\frac{1}{n}\sum_{i \in [n]}\|\nabla f_i(x) - \nabla F(x)\|^2 \leq 2A(F(x) - F_\star) + B.$$

We adopt a few extra assumptions than the previously stated ones for biased compressors. First, we consider a $(C, \zeta^2)$ bounded similarity assumption on the variance of the gradients among the workers in Assumption 7. This Assumption is stronger than Assumption 6, but a similar assumption was proposed in Sahu et al. (2021), and we rely on it for algebraic purposes in proving the convergence for the general case.

**Assumption 7.** *(**$(C, \zeta^2)$ bounded similarity**) The variance of gradients among workers is bounded, i.e., there exist constants, $C, \zeta \geq 0$ such that, $\frac{1}{n}\sum_{i \in [n]}\|\nabla f_i(x) - \nabla F(x)\|^2 \leq C\|\nabla F(x)\|^2 + \zeta^2$, for all $x \in \mathbb{R}^d$.*

We impose an extra assumption on the expected direction of the compressed gradient for biased compressors. A similar assumption was made in Dutta et al. (2020) and several classic biased compressors, such as Top-$k$, follow it.

**Assumption 8.** *(**Descent property of the compressed stochastic gradient**) Let $\mathcal{C}_t$ be a biased $\delta$-compressor such that $\frac{1}{n}\sum_{i=1}^{n}\mathcal{C}_t(g_{i,t}) = \tilde{g}$. There exists $0 < \alpha \leq 2$ and $\beta > 0$ such that*

$$\mathbb{E}\left[\tilde{g}^\top \nabla F | \nabla F\right] \geq \beta \mathbb{E}\|\nabla F\|^\alpha - R,$$

*where $R$ is a small scalar residual that may appear due to the numerical inexactness of some operators or other computational overheads.*

**Remark 3.** *The above assumption is general, and one can characterize many compressors with this. For instance, for Top-k, we have $\alpha = 2$, $\beta = k/d$ and $R = 0$. For simplicity and without loss of generality, one can consider $\alpha = 2$, $\beta = 1$, and $R = 0$.*

**Assumption 9.** *There exists $G \geq 0$ such that, $\|\nabla F_t\| \leq G$, for all $t \in [T]$.*

### B.1.1 INEQUALITIES USED

1. If $a, b \in \mathbb{R}^d$ then we use a relaxed version of Peter-Paul inequality:

$$\|a + b\|^2 \le 2\|a\|^2 + 2\|b\|^2. \tag{4}$$

2. If $a, b \in \mathbb{R}^d$ then the following holds:

$$2\langle a, b \rangle \le 2\|a\|^2 + \frac{1}{2}\|b\|^2. \tag{5}$$

3. For $x_1, \ldots, x_n \in \mathbb{R}^d$ we have:

$$\|\sum_{i=1}^{n} x_i\|^2 \le n \sum_{i=1}^{n} \|x_i\|^2. \tag{6}$$

4. If $X$ is a random variable then:

$$\mathbb{E}\|X\|^2 = \|\mathbb{E}[X]\|^2 + \mathbb{E}[\|X - E[X]\|^2]. \tag{7}$$

**Lemma 4.** *Let $\mathcal{C}(\cdot) : \mathbb{R}^d \to \mathbb{R}^d$ be a $\delta$-compressor.*

*(i) If $\mathcal{C}(g)$ is unbiased then We have $\mathbb{E}\|\mathcal{C}(g)\|^2 \le (2 - \delta)\|g\|^2$.*

*(ii) If $\mathcal{C}(g)$ is biased then $\mathbb{E}\|\mathcal{C}(g)\|^2 \le 2(2 - \delta)\|g\|^2$.*

*Proof.* ($i$) Recall for unbiased $\delta$-compressors, we have $\mathbb{E}\|g - \mathcal{C}(g)\|^2 \le (1 - \delta)\|g\|^2$. Since $\mathbb{E}(\mathcal{C}(g)) = g$, from equation 7 we have,

$$\mathbb{E}\|\mathcal{C}(g)\|^2 \overset{\text{By } equation\ 7}{=} \mathbb{E}\|g - \mathcal{C}(g)\|^2 + \|g\|^2 \le (1 - \delta)\|g\|^2 + \|g\|^2 = (2 - \delta)\|g\|^2.$$

($ii$) On the other hand, for biased $\delta$-compressors, we have,

$$\mathbb{E}\|\mathcal{C}(g)\|^2 = \mathbb{E}\|g - g + \mathcal{C}(g)\|^2 \overset{\text{By } equation\ 4}{\le} 2\mathbb{E}\|g - \mathcal{C}(g)\|^2 + 2\|g\|^2 \le 2(1-\delta)\|g\|^2 + 2\|g\|^2 = 2(2-\delta)\|g\|^2.$$

$\square$

**Lemma 5.** *Let $F$ follow Assumption 6. Then we have for all $t \ge 0$,*

$$\frac{1}{n} \sum_{i=1}^{n} \|\nabla f_{i,t}\|^2 \le 2A(F_t - F_\star) + B + \|\nabla F_t\|^2. \tag{8}$$

*Proof.* The proof follows from the fact that $\frac{1}{n} \sum_{i=1}^{n} \|\nabla f_{i,t}\|^2 = \frac{1}{n} \sum_{i=1}^{n} \|\nabla f_{i,t} - \nabla F_t + \nabla F_t\|^2$ and $F_t := \frac{1}{n} \sum_{i=1}^{n} f_{i,t}$ for all $t \ge 0$. Therefore,

$$\frac{1}{n} \sum_{i=1}^{n} \|\nabla f_{i,t}\|^2 \quad = \quad \frac{1}{n} \sum_{i=1}^{n} \|\nabla f_{i,t} - \nabla F_t + \nabla F_t\|^2$$

$$= \quad \frac{1}{n} \sum_{i=1}^{n} \|\nabla f_{i,t} - \nabla F_t\|^2 + \|\nabla F_t\|^2$$

$$\overset{\text{By Assumption 6}}{\le} \quad 2A(F_t - F_\star) + B + \|\nabla F_t\|^2.$$

Hence the result. $\square$

## B.2 CONVERGENCE OF GD

This section provides the convergence proofs GD on strongly convex and nonconvex functions with PL conditions as given in Lemma 1 and Lemma 2.

### B.2.1 CONVERGENCE OF GD ON STRONGLY CONVEX FUNCTIONS

**Lemma 1.** *(**Gradient descent with unbiased compressor**) Let $F$ follow Assumptions 1 and 2. Then with stepsize $\eta \leq \frac{1}{(2-\delta_t)L}$, the sequence of iterates, $\{x_t\}_{t\geq 0}$ of compressed GD updates satisfy*

$$E_{C_t}(\|x_{t+1} - x_\star\|^2) \leq \left(1 - 2\mu\eta + \eta^2\mu L(2 - \delta_t)\right)\|x_t - x_\star\|^2. \tag{9}$$

*Proof.* From the GD update in equation 2, we have

$$x_{t+1} - x_\star = x_t - x_\star - \eta\mathcal{C}_t(\nabla F(x_t)).$$

Squaring both sides and expanding, we have

$$\|x_{t+1} - x_\star\|^2 = \|x_t - x_\star\|^2 - 2\eta\mathcal{C}_t\left(\nabla F_t\right)^T\left(x_t - x_\star\right) + \eta^2\|\mathcal{C}_t(\nabla F_t)\|^2.$$

By taking expectation on the randomness of the compressors $\mathcal{C}_t$ we get:

$$
\begin{aligned}
\mathbb{E}_{\mathcal{C}_t}\left(\|x_{t+1} - x_\star\|^2\right) &= & \|x_t - x_\star\|^2 - 2\eta\nabla F_t^T(x_t - x_\star) + \eta^2\mathbb{E}_{\mathcal{C}_t}\|C_t(\nabla F_t)\|^2 \\
&\overset{\text{By Assumption 2}}{\leq} & \|x_t - x_\star\|^2 + 2\eta\left(F_\star - F_t\right) - \mu\eta\|x_t - x_\star\|^2 \\
& & +\eta^2(2 - \delta_t)\|\nabla F_t\|^2 \\
&\overset{\text{By Assumption 1}}{\leq} & \|x_t - x_\star\|^2 + 2\eta\left(F_\star - F_t\right) - \mu\eta\|x_t - x_\star\|^2 \\
& & +2\eta^2 L(2 - \delta_t)(F_t - F_\star) \\
&\leq & (1 - \mu\eta)\|x_t - x_\star\|^2 + 2\eta\left(\eta L(2 - \delta_t) - 1\right)(F_t - F_\star) \\
&\overset{\text{By Assumption 2}}{\leq} & (1 - \mu\eta)\|x_t - x_*\|^2 + \mu\eta\left(\eta L(2 - \delta_t) - 1\right)\|x_t - x_*\|^2 \\
&\leq & \left(1 - 2\mu\eta + \eta^2\mu L(2 - \delta_t)\right)\|x_t - x_\star\|^2.
\end{aligned}
$$

This completes the proof. □

### B.2.2 CONVERGENCE OF GD ON NONCONVEX FUNCTIONS WITH PL CONDITION

**Lemma 2.** *(**Gradient descent with unbiased compressor**) Let $F$ follow Assumptions 1 and 4. Then with stepsize $\eta = \frac{1}{L}$, the sequence of iterates, $\{x_t\}_{t\geq 0}$ of compressed GD updates satisfy*

$$E_{C_t}(F_{t+1}) - F_\star \leq \left(1 - \frac{\delta_t\mu}{L}\right)(F_t - F_\star). \tag{10}$$

*Proof.* Using the $L$-smoothness of $F$ as in Assumption 1 we have

$$
\begin{aligned}
F_{t+1} &\leq & F_t + \langle\nabla F_t, x_{t+1} - x_t\rangle + \frac{L}{2}\|x_{t+1} - x_t\|^2 \\
&\overset{\text{By } equation\ 2}{\leq} & F_t - \eta\langle\nabla F_t, \mathcal{C}_t(\nabla F(x_t))\rangle + \frac{\eta^2 L}{2}\|\mathcal{C}_t(\nabla F(x_t))\|^2.
\end{aligned}
$$

By taking the expectation on the randomness of $\mathcal{C}_t$ and by using the GD updates from equation 2, we have

$$
\begin{aligned}
\mathbb{E}_{\mathcal{C}_t}(F_{t+1}) &\leq & F_t - \frac{1}{L}\|\nabla F_t\|^2 + \frac{1}{2L}\mathbb{E}_{\mathcal{C}_t}\|\mathcal{C}_t(\nabla(F_t))\|^2 \\
&\overset{\text{By Lemma 4}}{\leq} & F_t - \left(\frac{1}{L} - \frac{2 - \delta_t}{2L}\right)\|\nabla F_t\|^2 \\
&\leq & F_t - \frac{\delta_t}{2L}\|\nabla F_t\|^2 \\
&\overset{\text{By Assumption 4}}{\leq} & F_t - \frac{\delta_t}{2L}2\mu(F_t - F_\star).
\end{aligned}
$$

Finally, subtracting $F_\star$ from both sides, we get

$$\mathbb{E}_{\mathcal{C}_t}(F_{t+1}) - F_\star \leq \left(1 - \frac{\delta_t}{L}\mu\right)(F_t - F_\star).$$

This completes the proof. □

### B.3 Convergence proofs for nonconvex distributed SGD

In this section, we provide the convergence proofs of compressed distributed SGD on nonconvex functions. We start with the key inequalities used in our proofs.

#### B.3.1 Unbiased compressors

**Lemma 3.** *(Compression variance) Let $\mathcal{C}_t$ be unbiased $\delta_t$-compressor for all $t \in [T]$, and let $F$ follow Assumption 6, and the stochastic noise follow Assumption 5. Then we have*

$$\mathbb{E}\left[\left\|\frac{1}{n}\left(\sum_{i=1}^{n}\mathcal{C}_t(g_{i,t}) - \sum_{i=1}^{n}\nabla f_{i,t}\right)\right\|^2 \Big| x_t\right] \leq \tag{11}$$

$$\frac{1}{n}\big((1-\delta_t)(M+1) + M\big)\big(2A(F_t - F_\star) + B + \|\nabla F_t\|^2\big) + \frac{(2-\delta_t)\sigma^2}{n}.$$

*Proof.* We note that the compression operator, $\mathcal{C}_t$, and the stochastic noise, $\xi_{i,t}$, are independent. Therefore, while taking expectation on the randomness of the compression operator, $\mathcal{C}_t$, we condition on the other source of randomness, and vice versa. We use $\mathbb{E}_{\mathcal{C}_t}$ to denote the expectation taken on the randomness of the compression operator, $\mathcal{C}_t$, and conditioned on other sources of randomness. So, taking expectation on the randomness of the compression operator, $\mathcal{C}_t$ we have

$$\mathbb{E}_{\mathcal{C}_t}\left\|\frac{1}{n}\left(\sum_{i=1}^{n}\mathcal{C}_t(g_{i,t}) - \sum_{i=1}^{n}\nabla f_{i,t}\right)\right\|^2$$

$$\overset{\mathbb{E}_{\mathcal{C}_t}(\mathcal{C}_t(g_{i,t}))=g_{i,t}}{=} \frac{1}{n^2}\sum_{i=1}^{n}\mathbb{E}_{\mathcal{C}_t}\|\mathcal{C}_t(g_{i,t}) - \nabla f_{i,t}\|^2 + \frac{2}{n^2}\sum_{i\neq j}\langle g_{i,t} - \nabla f_{i,t}, g_{j,t} - \nabla f_{j,t}\rangle$$

$$\overset{g_{i,t}=\nabla f_{i,t}+\xi_{i,t}}{=} \frac{1}{n^2}\sum_{i=1}^{n}\mathbb{E}_{\mathcal{C}_t}\|\mathcal{C}_t(g_{i,t}) - g_{i,t} + \xi_{i,t}\|^2 + \frac{2}{n^2}\sum_{i\neq j}\langle g_{i,t} - \nabla f_{i,t}, g_{j,t} - \nabla f_{j,t}\rangle$$

$$\overset{\mathbb{E}_{\mathcal{C}_t}(\mathcal{C}_t(g_{i,t}))=g_{i,t}}{=} \frac{1}{n^2}\sum_{i=1}^{n}\Big(\mathbb{E}_{\mathcal{C}_t}\|\mathcal{C}_t(g_{i,t}) - g_{i,t}\|^2 + \mathbb{E}_{\mathcal{C}_t}\|\xi_{i,t}\|^2\Big) + \frac{2}{n^2}\sum_{i\neq j}\langle g_{i,t} - \nabla f_{i,t}, g_{j,t} - \nabla f_{j,t}\rangle$$

$$\leq \frac{1}{n^2}\sum_{i=1}^{n}\big((1-\delta_t)\|g_{i,t}\|^2 + \|\xi_{i,t}\|^2\big) + \frac{2}{n^2}\sum_{i\neq j}\langle g_{i,t} - \nabla f_{i,t}, g_{j,t} - \nabla f_{j,t}\rangle.$$

Taking expectation conditioned on $x_t$, and by using the tower property of expectation, we get

$$\mathbb{E}\left[\mathbb{E}_{\mathcal{C}_t}\left[\left\|\frac{1}{n}\left(\sum_{i=1}^{n}\mathcal{C}_t(g_{i,t}) - \sum_{i=1}^{n}\nabla f_{i,t}\right)\right\|^2\right]\Big| x_t\right] \leq \frac{1}{n^2}\sum_{i=1}^{n}\big((1-\delta_t)\mathbb{E}[\|g_{i,t}\|^2|x_t] + \mathbb{E}[\|\xi_{i,t}\|^2|x_t]\big).$$

The equality holds as $\mathbb{E}(g_{i,t}|x_t) = \nabla f_{i,t}$ and $\mathbb{E}(g_{j,t}|x_t) = \nabla f_{j,t}$, for all $i \neq j, i, j \in [n]$. By using Assumption 5, write the above expression as

$$\frac{1}{n^2}\sum_{i=1}^{n}\big((1-\delta_t)\mathbb{E}[\|g_{i,t}\|^2|x_t] + \mathbb{E}[\|\xi_{i,t}\|^2|x_t]\big)$$

$$\leq \frac{1}{n^2}\sum_{i=1}^{n}\big((1-\delta_t)(M+1)\|\nabla f_{i,t}\|^2 + (1-\delta_t)\sigma^2 + M\|\nabla f_{i,t}\|^2 + \sigma^2\big)$$

$$\overset{\text{By Lemma 5}}{\leq} \frac{1}{n}\big((1-\delta_t)(M+1) + M\big)\big(2A(F_t - F_\star) + B + \|\nabla F_t\|^2\big) + \frac{1}{n}(2-\delta_t)\sigma^2.$$

Hence the result. $\qquad\square$

Based on the previous Lemma, the next lemma quantifies the quantity $\mathbb{E}\left\|\frac{1}{n}\sum_{i=1}^{n}\mathcal{C}_t(g_{i,t})\right\|^2$.

**Lemma 6.** *Let $\mathcal{C}_t$ be unbiased $\delta_t$-compressor for all $t \in [T]$. Let $F$ follow Assumptions 3, 6, and the stochastic noise follow Assumption 5. Then*

$$\mathbb{E} \left\| \frac{1}{n} \sum_{i=1}^{n} \mathcal{C}_t(g_{i,t}) \right\|^2 \leq \frac{2A\beta_t}{n} (F_t - F_\star) + \left(1 + \frac{\beta_t}{n}\right) \|\nabla F_t\|^2 + \frac{B\beta_t}{n} + \left(\frac{2 - \delta_t}{n}\right) \sigma^2, \quad (12)$$

*where $\beta_t := (1 - \delta_t)(M + 1) + M$.*

*Proof.* Taking expectation on the randomness of the compression operator, $\mathcal{C}_t$, we have

$$\mathbb{E}_{\mathcal{C}_t} \left\| \frac{1}{n} \sum_{i=1}^{n} \mathcal{C}_t(g_{i,t}) \right\|^2 = \mathbb{E}_{\mathcal{C}_t} \| \frac{1}{n} \sum_{i=1}^{n} \mathcal{C}_t(g_{i,t}) - \nabla F_t + \nabla F_t \|^2$$

$$= \mathbb{E}_{\mathcal{C}_t} \left\| \frac{1}{n} \sum_{i=1}^{n} \mathcal{C}_t(g_{i,t}) - \nabla F_t \right\|^2 + \|\nabla F_t\|^2 + 2\langle \frac{1}{n} \sum_{i=1}^{n} g_{i,t} - \nabla F_t, \nabla F_t \rangle$$

$$\overset{\text{By Lemma 3}}{\leq} \frac{1}{n} \left((1 - \delta_t)(M + 1) + M\right) \left(2A(F_t - F_\star) + B + \|\nabla F_t\|^2\right) + \frac{1}{n}(2 - \delta_t)\sigma^2$$

$$+ \|\nabla F_t\|^2 + 2\langle \frac{1}{n} \sum_{i=1}^{n} g_{i,t} - \nabla F_t, \nabla F_t \rangle. \quad (13)$$

Finally, we note that $\mathbb{E}(g_{i,t}|x_t) = f_{i,t}$. By using the tower property of expectation, we denote $\mathbb{E}\|\frac{1}{n} \sum_{i=1}^{n} \mathcal{C}_t(g_{i,t})\|^2 = \mathbb{E}(\mathbb{E}_{\mathcal{C}_t}\|\frac{1}{n} \sum_{i=1}^{n} \mathcal{C}_t(g_{i,t})\|^2 | x_t)$. Taken together, from equation 13, we have

$$\mathbb{E}\|\frac{1}{n} \sum_{i=1}^{n} \mathcal{C}_t(g_{i,t})\|^2$$

$$\leq \frac{1}{n} \left((1 - \delta_t)(M + 1) + M\right) \left(2A(F_t - F_\star) + B + \|\nabla F_t\|^2\right) + \frac{1}{n}(2 - \delta_t)\sigma^2 + \|\nabla F_t\|^2.$$

Hence the result. $\qquad \square$

Finally, we can quote the non-convex descent lemma for compressed distributed SGD.

**Lemma 7.** *(**Non-convex descent lemma for unbiased compressors**) Let Assumptions 1 , 5, and 6 hold, and let $\mathcal{C}_t$ be unbiased $\delta_t$-compressor for all $t \in [T]$. Then*

$$\mathbb{E}(F_{t+1}) - F_\star \leq \left(1 + \frac{AL\eta_t^2 \beta_t}{n}\right) (\mathbb{E}(F_t) - F_\star) - \eta_t \left(1 - \frac{L\eta_t}{2} - \frac{L\eta_t \beta_t}{n}\right) \mathbb{E}\|\nabla F_t\|^2$$

$$+ \frac{L\eta_t^2}{2} \left(\frac{B\beta_t}{n} + \left(\frac{2 - \delta_t}{n}\right)\sigma^2\right).$$

*Proof.* By using the $L$-smoothness of $F$ we have

$$F_{t+1} \leq F_t - \langle \nabla F_t, x_{t+1} - x_t \rangle + \frac{L}{2} \|x_{t+1} - x_t\|^2.$$

By using the update rule $x_{t+1} - x_t = -\frac{\eta_t}{n} \sum_{i=1}^{n} \mathcal{C}_t(g_{i,t})$ the above becomes

$$F_{t+1} \leq F_t - \langle \nabla F_t, \frac{\eta_t}{n} \sum_{i=1}^{n} \mathcal{C}_t(g_{i,t}) \rangle + \frac{L\eta_t^2}{2} \|\frac{1}{n} \sum_{i=1}^{n} \mathcal{C}_t(g_{i,t})\|^2. \quad (14)$$

Taking expectation with respect to the randomness of $\mathcal{C}_t$ on the above expression for all $t \in [T]$, we find

$$\mathbb{E}_{\mathcal{C}_t}(F_{t+1}) \leq F_t - \langle \nabla F_t, \frac{\eta_t}{n} \sum_{i=1}^{n} g_{i,t} \rangle + \frac{L\eta_t^2}{2} \mathbb{E}_{\mathcal{C}_t}\|\frac{1}{n} \sum_{i=1}^{n} \mathcal{C}_t(g_{i,t})\|^2.$$

Taking expectation conditioned on $x_t$, we have

$$\mathbb{E}(F_{t+1}|x_t) \leq \mathbb{E}(F_t|x_t) - \eta_t \mathbb{E}\|\nabla F_t\|^2 + \frac{L\eta_t^2}{2} \mathbb{E}\left(\|\frac{1}{n} \sum_{i=1}^{n} \mathcal{C}_t(g_{i,t})\|^2 | x_t\right).$$

By using Lemma 6 on the above, we find

$$
\begin{aligned}
\mathbb{E}(F_{t+1}|x_t) \leq\ & \mathbb{E}(F_t|x_t) - \eta_t\mathbb{E}\|\nabla F_t\|^2 \\
& + \frac{L\eta_t^2}{2}\left(\frac{2A\beta_t}{n}(F_t - F_\star) + \left(1 + \frac{\beta_t}{n}\right)\|\nabla F_t\|^2 + \frac{B\beta_t}{n} + \left(\frac{2-\delta_t}{n}\right)\sigma^2\right).
\end{aligned}
$$

Taking the final expectation, by using the tower property of expectation, and rearranging the terms, we have

$$
\begin{aligned}
\mathbb{E}(F_{t+1}) - F_\star \leq\ & \left(1 + \frac{AL\eta_t^2\beta_t}{n}\right)(\mathbb{E}(F_t) - F_\star) - \eta_t\left(1 - \frac{L\eta_t}{2} - \frac{L\eta_t\beta_t}{n}\right)\mathbb{E}\|\nabla F_t\|^2 \\
& + \frac{L\eta_t^2}{2}\left(\frac{B\beta_t}{n} + \left(\frac{2-\delta_t}{n}\right)\sigma^2\right).
\end{aligned}
\tag{15}
$$

Hence the result.

$\square$

### B.3.2 BIASED COMPRESSORS

**Lemma 8.** *Let $\mathcal{C}_t$ be biased $\delta$-compressors for all $t \in [T]$, and let $F$ follow Assumption 7, and the stochastic noise follow Assumption 5. Then we have*

$$
\mathbb{E}\left[\|\tfrac{1}{n}\sum_{i=1}^n \mathcal{C}_t(g_{i,t})\|^2|x_t\right] \leq 2(2-\delta_t)(M+1)(C+1)\|\nabla F_t\|^2 + 2(2-\delta_t)\left((M+1)\zeta^2 + \sigma^2\right).
$$

*Proof.* Taking expectation, $\mathbb{E}_{\mathcal{C}_t}$ on $\|\frac{1}{n}\sum_{i=1}^n \mathcal{C}_t(g_{i,t})\|^2$ we have

$$
\mathbb{E}_{\mathcal{C}_t}\|\tfrac{1}{n}\sum_{i=1}^n \mathcal{C}_t(g_{i,t})\|^2
$$

$$
\overset{\text{By } equation\ 6}{\leq} \frac{1}{n}\sum_{i=1}^n \mathbb{E}_{\mathcal{C}_t}\|\mathcal{C}_t(g_{i,t})\|^2
$$

$$
\overset{\text{By Lemma 4}}{\leq} 2(2-\delta_t)\frac{1}{n}\sum_{i=1}^n \|g_{i,t}\|^2
$$

$$
\overset{\text{By Assumption 5}}{\leq} 2(2-\delta_t)(M+1)\frac{1}{n}\sum_{i=1}^n\|\nabla f_{i,t}\|^2 + 2(2-\delta_t)\sigma^2
$$

$$
= 2(2-\delta_t)(M+1)\frac{1}{n}\sum_{i=1}^n\|\nabla f_{i,t} - \nabla F_t\|^2 + 2(2-\delta_t)(M+1)\|\nabla F_t\|^2 + 2(2-\delta_t)\sigma^2
$$

$$
\overset{\text{By Assumption 7}}{\leq} 2(2-\delta_t)(M+1)(C\|\nabla F_t\|^2 + \zeta^2) + 2(2-\delta_t)(M+1)\|\nabla F_t\|^2 + 2(2-\delta_t)\sigma^2.
$$

Now, by taking the conditional expectation on $x_t$ and using Lemma 5, we obtain the result.

$\square$

**Lemma 9.** *(Non-convex descent lemma for biased compressors) Let Assumptions 1, 5, and 7 hold, and let $\mathcal{C}_t$ be biased $\delta$-compressor for all $t \in [T]$ that follows Assumption 8. Then*

$$
\begin{aligned}
\eta_t\left(\beta\mathbb{E}\|\nabla F_t\|^\alpha - L\eta_t(2-\delta_t)(M+1)(C+1)\mathbb{E}\|\nabla F_t\|^2\right) \leq\ & \mathbb{E}(F_t - F_{t+1}) \\
& + L\eta_t^2(2-\delta_t)\left(\sigma^2 + (M+1)\zeta^2\right) + \eta_t R.
\end{aligned}
$$

*Proof.* By using the $L$-smoothness of $F$, and using the update rule $x_{t+1} - x_t = -\frac{\eta_t}{n}\sum_{i=1}^n \mathcal{C}_t(g_{i,t})$ we have from equation 14:

$$
F_{t+1} \leq F_t - \langle\nabla F_t, \tfrac{\eta_t}{n}\sum_{i=1}^n \mathcal{C}_t(g_{i,t})\rangle + \frac{L\eta_t^2}{2}\|\tfrac{1}{n}\sum_{i=1}^n \mathcal{C}_t(g_{i,t})\|^2.
$$

Taking expectation with respect to the randomness of $\nabla F_t$ on the above expression for all $t \in [T]$ and by using Assumption 8, we find

$$\mathbb{E}(F_{t+1}) \leq F_t - \beta\eta_t\mathbb{E}\|\nabla F_t\|^\alpha + \eta_t R + \frac{L\eta_t^2}{2}\mathbb{E}\left(\|\tfrac{1}{n}\sum_{i=1}^n \mathcal{C}_t(g_{i,t})\|^2|\nabla F_t\right).$$

Taking expectation conditioned on $x_t$, we have

$$\mathbb{E}(F_{t+1}|x_t) \leq F_t - \beta\eta_t\mathbb{E}\|\nabla F_t\|^\alpha + \eta_t R + \frac{L\eta_t^2}{2}\mathbb{E}\left[\|\tfrac{1}{n}\sum_{i=1}^n \mathcal{C}_t(g_{i,t})\|^2|x_t\right].$$

Now using Lemma 8 we get
$$\begin{aligned}
\mathbb{E}(F_{t+1}|x_t) \leq\ & F_t - \beta\eta_t\mathbb{E}\|\nabla F_t\|^\alpha + \eta_t R \\
& + L\eta_t^2\left((2-\delta_t)(M+1)(1+C)\|\nabla F_t\|^2 + (2-\delta_t)\left((M+1)\zeta^2 + \sigma^2\right)\right).
\end{aligned}$$

Taking the final expectation, by using the tower property of expectation, and rearranging the terms, we have the result. $\qquad\square$

NONCONVEX CONVERGENCE RESULTS

The next Lemma is instrumental in proving the nonconvex convergence of distributed SGD with $\delta$-compressors.

**Lemma 10.** *Mishchenko et al. (2020) Let for $0 \leq t \leq T$ the following holds:*
$$p_{t+1} \leq (1+a)p_t - bq_t + c, \tag{16}$$
*where $\{p_t\}_{t=0}^T$ and $\{q_t\}_{t=0}^T$ are non-negative sequences and $a, b, c \geq 0$ are constants. Then*
$$\min_{t=0,1,\cdots T-1} q_t \leq \frac{(1+a)^T}{bT}p_0 + \frac{c}{b}. \tag{17}$$

*Proof.* Dividing both sides of equation 16 by $(1+a)^{t+1}$ and summing from $t = 0, 1, \cdots, T$ we have
$$\sum_{t=0}^T \frac{1}{(1+a)^{t+1}}p_{t+1} \leq \sum_{t=0}^T \frac{1}{(1+a)^t}p_t - \sum_{t=0}^T \frac{b}{(1+a)^{t+1}}q_t + \sum_{t=0}^T \frac{c}{(1+a)^{t+1}},$$
which after rearranging is
$$\sum_{t=0}^t \frac{b}{(1+a)^{t+1}}q_t \leq p_0 - \frac{1}{(1+a)^{T+1}}p_{T+1} + \sum_{t=0}^T \frac{c}{(1+a)^{t+1}}.$$
Noting $\sum_{t=0}^T \frac{1}{(1+a)^{t+1}} \leq \frac{1}{1-\frac{1}{1+a}} - 1 = \frac{1}{a}$, we have
$$\min_{t=0,1,\cdots T} q_t \sum_{t=0}^T \frac{1}{(1+a)^{t+1}} \leq \sum_{t=0}^T \frac{1}{(1+a)^{t+1}}q_t \leq \frac{p_0}{b} + \frac{c}{ab}. \tag{18}$$
Hence the result. $\qquad\square$

Finally, we are set to prove Theorem 1.

**Theorem 1.** *(Nonconvex convergence) (i) (Unbiased) Let Assumptions 1, 5, and 6 hold, and let $\mathcal{C}_t$ be unbiased $\delta_t$-compressor for all $t \in [T]$. For a fixed stepsize $\eta_t := \eta \leq \min\left(\frac{1}{\frac{L}{2}+\frac{L(2M+1)}{n}}, \left(\frac{AL(2M+1)T}{n}\right)^{-\frac{1}{2}}\right)$ we have:*

$$\min_{t=0,1,\cdots T-1} \mathbb{E}\|\nabla F(x_t)\|^2 \leq \frac{3}{T\eta\left(1-\frac{L\eta}{2}-\frac{L\eta(2M+1)}{n}\right)}(F_0 - F_\star) + \frac{L\eta\left(B(2M+1)+2\sigma^2\right)}{2n\left(1-\frac{L\eta}{2}-\frac{L\eta(2M+1)}{n}\right)}.$$

*(ii) (Biased) Let Assumptions 1, 3, 5, 7 and 9 hold, and let $\mathcal{C}_t$ be biased $\delta$-compressors for all $t \in [T]$ that follow Assumption 8. For a stepsize $\eta < \frac{\beta}{L(2-\delta_t)(M+1)(C+1)G^{2-\alpha}}$, we have:*

$\min_{t=0,1,\cdots T-1} \mathbb{E}\|\nabla F\|^\alpha \leq \frac{(F_0-F_\star)}{T\eta\beta\mathcal{A}} + \hat{\sigma}$, where $\hat{\sigma} = \frac{2L\eta\left(\sigma^2+(M+1)\zeta^2\right)}{\beta\mathcal{A}} + \frac{R}{\beta\mathcal{A}}$, and $\mathcal{A} := \left(1 - \frac{L\eta(2-\delta_t)(M+1)(C+1)G^{2-\alpha}}{\beta}\right)$.

*Proof.* (*i*) From Lemma 7 we have

$$
\mathbb{E}(F_{t+1}) - F_\star \leq \left(1 + \frac{AL\eta_t^2 \beta_t}{n}\right)(\mathbb{E}(F_t) - F_\star) - \eta_t\left(1 - \frac{L\eta_t}{2} - \frac{L\eta_t\beta_t}{n}\right)\mathbb{E}\|\nabla F_t\|^2
$$
$$
+ \frac{L\eta_t^2}{2}\left(\frac{B\beta_t}{n} + \left(\frac{2-\delta_t}{n}\right)\sigma^2\right).
$$

The above inequality satisfies the condition of equation 16 with $a = \frac{AL\eta^2(2M+1)}{n}, b = \eta\left(1 - \frac{L\eta}{2} - \frac{L\eta(2M+1)}{n}\right), c = \frac{L\eta^2}{2}\left(\frac{B(2M+1)}{n} + \frac{2\sigma^2}{n}\right)$. Therefore, we obtain

$$
\min_{t=0,1,\cdots T-1}\mathbb{E}\|\nabla F(x_t)\|^2 \leq \frac{\left(1 + \frac{AL\eta^2(2M+1)}{n}\right)^T}{T\eta\left(1 - \frac{L\eta}{2} - \frac{L\eta(2M+1)}{n}\right)}(F_0 - F_\star) + \frac{\frac{L\eta^2}{2}\left(\frac{B(2M+1)}{n} + \frac{2\sigma^2}{n}\right)}{\eta\left(1 - \frac{L\eta}{2} - \frac{L\eta(2M+1)}{n}\right)}. \tag{19}
$$

Using that $x+1 \leq \exp x$ and with $\eta \leq \left(\frac{AL(2M+1)T}{n}\right)^{-\frac{1}{2}}$ in the first term of the RHS of equation 19, we get

$$
\left(1 + \frac{AL\eta^2(2M+1)}{n}\right)^T \leq \exp\left(\frac{AL\eta^2(2M+1)T}{n}\right) \leq \exp(1) \leq 3.
$$

Finally, using the above in the inequality (19), we have

$$
\min_{t=0,1,\cdots T-1}\mathbb{E}\|\nabla F(x_t)\|^2 \leq \frac{3}{T\eta\left(1 - \frac{L\eta}{2} - \frac{L\eta(2M+1)}{n}\right)}(F_0 - F_\star) + \frac{L\eta\left(B(2M+1) + 2\sigma^2\right)}{2n\left(1 - \frac{L\eta}{2} - \frac{L\eta(2M+1)}{n}\right)}.
$$

Hence the result.

(*ii*) With $\|\nabla F_t\| \leq G$ from Assumption 9, we have

$$
\eta_t\beta\mathbb{E}\|\nabla F_t\|^\alpha\left(1 - \frac{L\eta_t(2-\delta_t)(M+1)(C+1)G^{2-\alpha}}{\beta}\right) \leq \mathbb{E}(F_t - F_{t+1}) + L\eta_t^2(2-\delta_t)\left(\sigma^2 + (M+1)\zeta^2\right) + \eta_t R.
$$

Consider $\eta_t = \eta < \frac{\beta}{L(2-\delta_t)(M+1)(C+1)G^{2-\alpha}}$. Then the above inequality reduces to

$$
\mathbb{E}\|\nabla F_t\|^\alpha \leq \frac{\mathbb{E}(F_t - F_{t+1})}{\eta\beta\mathcal{A}} + \frac{L\eta(2-\delta_t)\left(\sigma^2 + (M+1)\zeta^2\right)}{\beta\mathcal{A}} + \frac{R}{\beta\mathcal{A}}.
$$

By unrolling the recurrence relation, we have

$$
\frac{1}{T}\sum_{t=0}^{T-1}\mathbb{E}\|\nabla F_t\|^\alpha \leq \frac{F_0 - \mathbb{E}(F_t)}{T\eta\beta\mathcal{A}} + \frac{L\eta\left(\sigma^2 + (M+1)\zeta^2\right)}{T\beta\mathcal{A}}\sum_{t=0}^{T-1}(2-\delta_t) + \frac{R}{\beta\mathcal{A}}.
$$

With $0 \leq \delta_t \leq 1$ we get (also, using Assumption 3),

$$
\frac{1}{T}\sum_{t=0}^{T-1}\mathbb{E}\|\nabla F_t\|^\alpha \leq \frac{F_0 - F_\star}{T\eta\beta\mathcal{A}} + \frac{2L\eta\left(\sigma^2 + (M+1)\zeta^2\right)}{\beta\mathcal{A}} + \frac{R}{\beta\mathcal{A}},
$$

which further reduces to

$$
\min_{t=0,1,\cdots,T-1}\mathbb{E}\|\nabla F_t\|^\alpha \leq \frac{F_0 - F_\star}{T\eta\beta\mathcal{A}} + \frac{2L\eta\left(\sigma^2 + (M+1)\zeta^2\right)}{\beta\mathcal{A}} + \frac{R}{\beta\mathcal{A}}.
$$

Hence the proof.

□

## C  VARIANTS OF LEGACY

To simplify the selection of hyperparameters used in LEGACY ($\{\lambda_i\}_{i=1}^p$ for thresholds and $\{\delta_i\}_{i=1}^p$ for compression levels), we propose a simplified version of LEGACY in §C.1 called **Simple**-LEGACY or S-LEGACY.

### C.1 SIMPLE LEGACY

We note that there is no recipe for choosing compression parameters. The choice of compression parameters depends on multiple factors such as the dataset used, DNN model architecture, network topology, network bandwidth, and many more; see Xu et al. (2021a) and references therein. In contrast to compute-heavy state-of-the-art adaptive compressors, LEGACY is based on two simple propositions: (a) the layer size of the DNNs influences in choosing how much one needs to compress, smaller layers have insignificant effect compared to large layers, and (b) the training phase of the DNNs can be a critical contributor in the adaptive compressor design, the end training phase can tolerate severe compression without any accuracy lost. While the layer sizes can be determined and grouped based on their relative sizes, the only rule for choosing compression parameters based on the training phase is to *choose to decrease compression parameters over iterations*; §5.4 validates this on QSGD and PowerSGD.

To simplify the selection of hyperparameters used in LEGACY ($\{\lambda_i\}_{i=1}^p$ for thresholds and $\{\delta_i\}_{i=1}^p$ for compression levels), we proposed a simplified version called S-LEGACY. This version requires only two hyperparameters for the epoch or layer-based approach and three for the mixed approach. That is, S-LEGACY-E requires only a default compression parameter $\delta_u$ and the number of training phases $n$; S-LEGACY-L requires $\delta_u$ and a decrease ratio $s$; S-LEGACY-M (stands for using both layer and epoch-based) requires $\delta_u$, $n$, and $s$.

S-LEGACY determines grouping and compression parameters based on the specified hyperparameters and two additional functions that depend on the compressors used during training: (*i*) $vol(\delta)$ computes or estimates the communicated data volume $v$ for a given compression parameter $\delta$; and (*ii*) $vol^{-1}(v)$ determines or estimates the compression parameter $\delta$ that produces a specified data volume $v$.

**Grouping.** In LEGACY, we used $\{\lambda_i\}_{i=1}^p$ to define training phases or layer groups. In S-LEGACY, we simplify this grouping as follows:

**S-LEGACY-E.** The training duration $T$ is uniformly divided into $n$ phases. Group $g_i$ consists of iterations within the interval $\left((i-1)\frac{T}{n}, i\frac{T}{n}\right]$.

**S-LEGACY-L.** Layers are grouped by order of magnitude. Group $g_i$ consists of layers L satisfying $100^{i-1} \leq |L| < 100^i$, where $|L|$ denotes the layer size.

**Compression parameters.** S-LEGACY eliminates the need to set the compression parameters for each group manually. Instead, these parameters are automatically calculated to ensure a similar communicated data volume as the default compression parameter $\delta_u$, while following the principles established by LEGACY.

**S-LEGACY-E.** First, compute the uniform data volume $V_u$ that would be communicated with $\delta_u$. For group $g_i$, the compression parameter is $\delta_i = vol^{-1}(V_i)$, where $V_i$ is the data volume $V_i = \left(1.5 - \frac{i-1}{n-1}\right) V_u$. This ensures progressively aggressive compression across training phases, starting at $1.5V_u$ (first phase) and ending at $0.5V_u$ (last phase).

**S-LEGACY-L.** Compress the largest layer group $g_p$ more aggressively than $\delta_u$: $\delta_p \approx vol_p^{-1}\left((1-s) \cdot vol_p(\delta_u)\right)$, where $s \leq 5\%$ is the decrease ratio, and $vol_p(\delta)$ represents the data volume of the group $g_p$ when compressed with $\delta$. Distribute the saved volume $s \cdot vol_p(\delta_u)$ uniformly across the other groups: $\delta_i = vol_i^{-1}\left(\frac{s}{p-1} \cdot vol_p(\delta_u) + vol_i(\delta_u)\right)$. Since the groups are ordered by magnitude, this adjustment applies lighter compression to smaller groups and progressively more aggressive compression to larger groups, ensuring that $\delta_u$ is not exceeded except for $g_p$.

### C.1.1 MIXED APPROACH

The mixed approach, S-LEGACY-M, combines S-LEGACY-L and S-LEGACY-E, requiring the previously defined parameters: the default compression parameter $\delta_u$, the number of training phases $n$, and the decrease ratio $s$. S-LEGACY-M begins by applying S-LEGACY-E to partition the training period, then uses S-LEGACY-L to determine compression parameters for each layer group, with the compression parameter $\delta_i$ from S-LEGACY-E for the current phase serving as the default compression parameter. Use the epoch-based method (S-LEGACY-E) to divide the training period into $n$ phases and compute $\{\delta_i\}_{i=1}^n$. Then for each phase $i$, use $\delta_i$ as the default compression parameter for the layer-based method (S-LEGACY-L) to calculate compression parameters for layer groups within that phase; see results in §C.1.2.

Table 5: Accuracy for S-LEGACY methods.

| Network | Method | Uniform | S-LEGACY-L | S-LEGACY-E | S-LEGACY-M |
|---|---|---|---|---|---|
| AlexNet | Top-k | 78.1 | 79.3 | 78.6 | 81.1 |
| | Random-k | 67.7 | 72.4 | 71.13 | 72.6 |
| ResNet9 | Top-k | 85.86 | 86.7 | 87.52 | 88.03 |
| | Random-k | 77.58 | 81.8 | 81.9 | 82.04 |
| ResNet18 | Top-k | 64.4 | 66.2 | 66.18 | 66.5 |
| | Random-k | 50.4 | 51.9 | 52.2 | 52.1 |

### C.1.2 NUMERICAL RESULTS OF S-LEGACY-M

We train AlexNet and ResNet9 on CIFAR-10 and ResNet18 on CIFAR-100 using two workers. We compare the uniform compression methods, Top-1% and Random-1%, with S-LEGACY approaches using $n = 5$ phases and a decrease ratio of $s = 5\%$.

The results in Table 5 demonstrate that the simplified S-LEGACY approaches outperform uniform compression methods and alleviate the burden of manually selecting the hyperparameters required by LEGACY.

### C.1.3 STRATEGY II: EXAMPLE

To illustrate this approach, consider using a sparsifier with a baseline sparsity of 1%. Instead of applying uniform compression, we divide layers into quartiles: Q1 (largest 25%) to Q4 (smallest 25%), and slightly reduce Q1's sparsity to 0.99%, reallocating the saved compression budget across the other quartiles. In AlexNet, this small adjustment enables Q2 to be compressed at 8.4%, Q3 at 37.6%, and Q4 at 74.2%. Similarly, in ResNet18, with Q1 compressed at 0.99%, Q4 receives a full 100%, meaning all gradients are transmitted uncompressed. Though the difference in Q1 is minimal, the resulting improvement for smaller layers is substantial, justifying the observed gains in model performance.

Another angle in understanding compression across layers is from the perspective of communicated data volume. Let us divide the layers into two groups: small and large. Let $S$ represent the total size of the small layers (e.g., layers with fewer than $10^4$ elements) and $L$ represent the total size of the large layers (e.g., layers with more than $10^4$ elements). We assume $S \ll L$. E.g., in our experiments, the ratio, $\frac{L}{S+L}$ is 0.9996, 0.987, and 0.997 for Transformer, ResNet-9, and AlexNet, respectively.

We aim to select $\delta_s$ and $\delta_l$ such that the overall data volume remains consistent with that obtained using a uniform compression, $\delta$, throughout the training. Let the compressed data volume when using a uniform compression $\delta$ be $V_{\text{uniform}}$ and layer-based LEGACY (when using $\delta_s$ on the small layers and $\delta_l$ on the large layers) be $V_{\text{dynamic}}$.

We have $V_{\text{uniform}} \propto \delta(L + S)$ and $V_{\text{dynamic}} \propto \delta_l L + \delta_s S$. Therefore, to keep the same overall data volume, $V_{\text{uniform}} \approx V_{\text{dynamic}}$, implies $\delta(L + S) \approx \delta_l L + \delta_s S$, that is, $\delta L \approx \delta_l L + (\delta_s - \delta)S$. Based on our assumption, $S \ll L$, together with $\delta_l < \delta \ll \delta_s \leq 1$, we obtain $\delta \approx \delta_l$, as $(\delta_s - \delta)\frac{S}{L} \to 0$. We do not have any explicit assumption on $\delta_s$, so we can choose it close to 1, that is, easy compression. We postulate, it is better to compress the large layers and leave the small layers uncompressed.

## D ADDENDUM TO EXPERIMENTAL EVALUATIONS

In this section, we provide additional experimental details and benchmarking results, which we were unable to discuss in the main paper due to limited space.

### D.1 REPRODUCIBILITY

We implement the sparsifiers in PyTorch. Tables 9, 11, 12, 13, and 14 provide the experimental details for each of the tasks. We used the default hyperparameters provided in the mentioned repositories for each task.

We postulated that LEGACY can be used conjointly with any compression techniques in designing its compute-free, adaptive counterpart. In this section, we provide additional experimental details and benchmarking results to demonstrate that LEGACY can be seamlessly integrated with other compression classes: sparsification (Random-k), quantization (QSGD Alistarh et al. (2017)), and

Table 6: Summary of the benchmarks and quality metrics used in this work.

| Task | Model | Dataset | Training parameters | Quality metric | Baseline quality | Optimizer |
|------|-------|---------|--------------------|----------------|------------------|-----------|
| Image Classification | AlexNet | CIFAR-10 | 2,255,296 | Accuracy | 84.99% | SGD Robbins & Monro (1951) |
| | ResNet9 | CIFAR-10 | 6,573,120 | Accuracy | 92.07% | SGD Robbins & Monro (1951) |
| | ResNet18 | CIFAR-100 | 11,220,132 | Accuracy | 73.43% | SGD-M Nesterov (2013) |
| | ResNet50 | ImageNet | 25,559,081 | Accuracy | 59.43% | SGD Robbins & Monro (1951) |
| Recommendation | NCF | Movielens-20m | 31,832,577 | HR@10 | 95.53% | ADAM Kingma & Ba (2015) |
| Language Modelling | Transformer-XL | WikiText-103 | 191,950,298 | Perplexity | 39.47 | LAMB You et al. (2020) |
| | GPT-2 small | OpenWebText | 124,373,760 | Validation loss | 2.85 | AdamW Loshchilov & Hutter (2019) |
| Federated Learning | ResNet18 | CIFAR-10 | 11,173,962 | Accuracy | 85.37% | SGD-M Nesterov (2013) |

Table 7: Dataset and training configuration.

| Dataset Name | Size | Workers used | Training Time (min) | Independent Runs Performed |
|--------------|------|--------------|---------------------|----------------------------|
| CIFAR10 Krizhevsky et al. (2009) | 160MB | 2 | 5 | 15 |
| CIFAR100 Krizhevsky et al. (2009) | 160MB | 2 | 20 | 15 |
| ImageNet Deng et al. (2009) | 140GB | 4 | 2100 | 1 |
| Movielens-20m Harper & Konstan (2015) | 190MB | 4 | 2 | 10 |
| WikiText-103 Merity et al. (2017) | 500MB | 4 | 190 | 4 |
| OpenWebText Gokaslan et al. (2019) | 40GB | 2 | 1440 | 1 |

low-rank factorization (PowerSGD Vogels et al. (2019)). These results, which we could not cover in detail in the main paper due to limited space, further validate LEGACY's versatility across different compression approaches.

## D.2 LEGACY ON DIFFERENT COMPRESSION CLASSES

In this Section, we show the efficacy of LEGACY on Random-$k$ as the base compressor.

### D.2.1 RANDOM-$k$ IN LEGACY AS BASE COMPRESSOR

Following the configuration described in Section 5, we provide additional tests, using the Random-$k$ as the base compressor in LEGACY. Figure 6 displays the accuracy versus relative average data volume throughout training for AlexNet, ResNet-9, and Transformer-XL.

In Table 8, we report the accuracy of ResNet-9 and AlexNet, including standard deviations obtained through independent runs using Top-$k$ and Random-$k$ as base compressors in LEGACY. Top-$k$ demonstrates superior performance relative to Random-$k$. The tests revealed comparable findings to those discussed in Subsection 5.1, further validating the importance of small layers and the initial training phase in improving compression efficiency.

Table 8: Comparison of average compression ratios vs. mean accuracy with standard deviation derived from 7 runs.

| Method | Compression ratio | ResNet9 Average ratio | Accuracy | AlexNet Average ratio | Accuracy |
|--------|-------------------|-----------------------|----------|----------------------|----------|
| Baseline | N/A | 100% | $92.07 \pm 0.13$ | 100% | $84.98 \pm 0.34$ |
| Topk | 0.1% | 0.1% | $75.72 \pm 1.07$ | 0.1% | $65.53 \pm 0.86$ |
| Topk-epoch | $B_{0.05}E_{0.15}$ | 0.1% | $73.65 \pm 0.16$ | 0.1% | $59.85 \pm 4.9$ |
| Topk-epoch | $B_{0.15}E_{0.05}$ | 0.1% | $79.18 \pm 0.26$ | 0.1% | $66.25 \pm 0.62$ |
| Topk-layer | $S_{10}L0.1$ | 0.12% | $\mathbf{82.94 \pm 0.79}$ | 0.13% | $\mathbf{70.27 \pm 0.91}$ |
| Randomk | 0.1% | 0.1% | $50.04 \pm 0.8$ | 0.1% | $43.58 \pm 0.45$ |
| Randomk-layer | $S_{10}L_{0.1}$ | 0.12% | $\mathbf{68.67 \pm 0.53}$ | 0.13% | $\mathbf{62.13 \pm 0.45}$ |

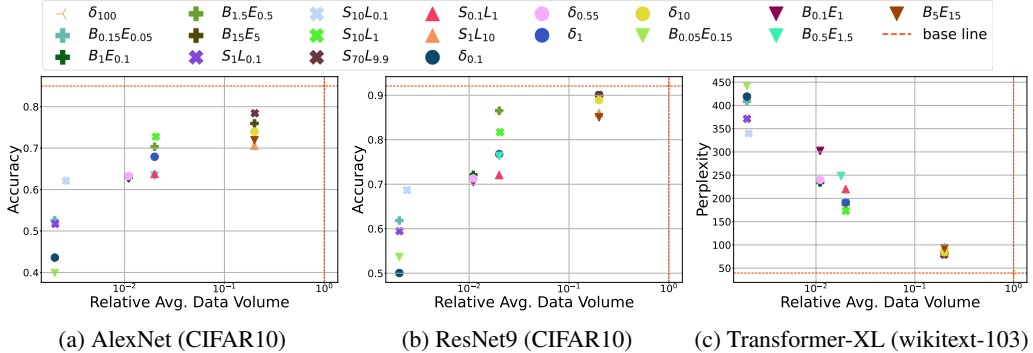

(a) AlexNet (CIFAR10)  (b) ResNet9 (CIFAR10)  (c) Transformer-XL (wikitext-103)

Figure 6: Layer-size and training epoch dependent Random-$k$ compression, where $S_{\delta_1}L_{\delta_2}$ means small layers($\leq 10^5$) compressed with compression factor, $\delta_1$ and large layers compressed with compression factor, $\delta_2$, and $B_{\delta_1}E_{\delta_2}$ denotes two-phase training, beginning phase (half of the total training epoch) with compression factor, $\delta_1$ and ending phase with compression factor $\delta_2$.

### D.3 SCALABILITY OF LEGACY

We performed our previous experiments on high-performance GPUs in a data center, connected by a fast network, and consisting of a limited number of workers. To evaluate performance in more constrained environments, we now simulate scenarios with a larger number of workers and restricted resources.

#### D.3.1 LEGACY ON CONSTRAINED ENVIRONMENTS

**Testbed and setup.** We trained ResNet-18 on CIFAR-10 using 50 workers, sharing a 1Gbps network bandwidth, with every worker operating on an Intel Xeon Platinum 8276 CPU instead of a GPU. In this part, we integrated error feedback (EF) in our tests; the implementation of EF is based on Sahu et al. (2021). We use Gloo `AllGather` for internodal communication. Figure 7 profiles the accuracy per wall clock time for 4100 seconds, which is the time required for compressors to complete 30 epochs. For the compression parameters of each method, we employed the following so that all methods transmit (almost) equal average data volume:

- Top-$k$: 1.7% uniform compression.
- Accordion: Set low and high compression ratio to $k_{low} = 0.1\%$ and $k_{high} = 10\%$, respectively, achieving an average compression ratio of 1.98%.
- Top-$k$ Epoch-based: The total training duration of 30 epochs was divided into four segments: three segments of 8 epochs each, followed by a final segment of 7 epochs. Compression ratios were set to 5%, 1%, 0.5%, and 0.1% for each segment, respectively, resulting in an average compression ratio of 1.75%.
- Top-$k$ Layer-based: Layers were categorized based on size into five groups: very small ($\leq$ 100), small ($\leq$ 600), medium ($\leq 10^5$), large ($\leq 10^6$), and very large ($\geq 10^6$). Assigned compression ratios were 80%, 50%, 20%, 5%, and 0.1% for each group respectively, transmitting 1.77% of the gradients.

**Results.** Although the no-compression baseline achieves the highest accuracy, the time required is also large in environments with limited and weak resources. In this test, the baseline needed more than 6 hours to complete 30 epochs, while the compression tests took $\approx 4100$ seconds, thereby achieving the best return for time. From Figure 7a, we can observe that the Epoch-based Top-$k$ strategy achieves the best performance in the first 1000 seconds, which is expected as the method is running through a light compression of 5% during this period, compared to the other compressors that are using around a 1.7% compression ratio. The uniform compressors required approximately double the time (a little less than $2000s$) to reach this level of accuracy. On the other hand, the Top-$k$ strategy based on layer size, stands out with the best accuracy when the layer size groups are more refined; creating more groups helps in controlling the compression for sensitive and small layers to achieve better accuracy.

**Takeaways.** In resource-limited environments, the strategies in LEGACY perform better in terms of obtaining a better accuracy faster. The initial mild compression phase of the epoch-based strategy

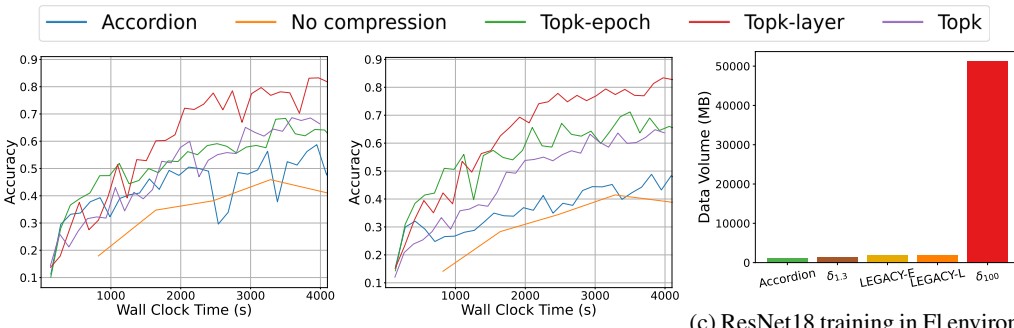

(a) ResNet18 training (with EF)  (b) ResNet18 training (without EF)  (c) ResNet18 training in Fl environment, total data volume.

Figure 7: In (a) and (b), we show accuracy vs. wall clock time of training ResNet-18 on CIFAR10, with and without EF, respectively. In (c), we show the total communicated data volume in ResNet-18 on CIFAR10 training in an FL environment; see legend in Figure 8.

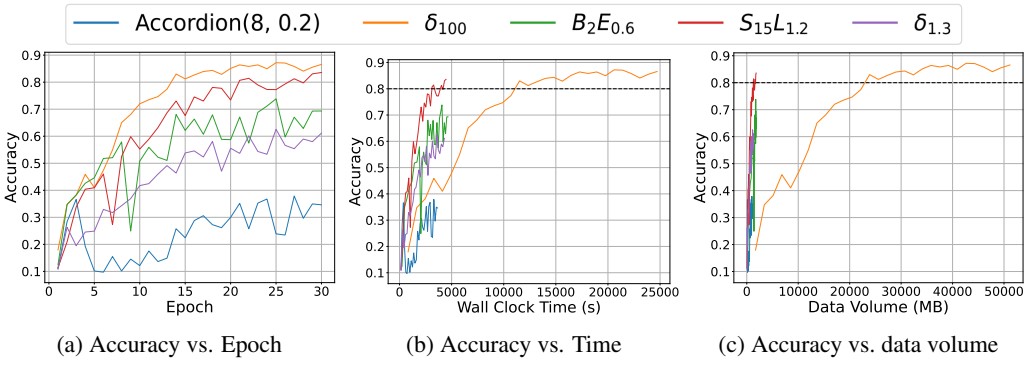

(a) Accuracy vs. Epoch  (b) Accuracy vs. Time  (c) Accuracy vs. data volume

Figure 8: Training ResNet18 on CIFAR10 in a FL Environment; $\delta_{100}$ is no compression baseline.

allows it to benefit from the early training phase and outperform other methods, which take significant time to match its performance, even after the epoch strategy enters the aggressive phase. On the other hand, applying light compression to small layers enhances model performance. In both strategies, creating more groups aids in refining the compression more effectively to achieve better performance.

### D.3.2   FEDERATED TRAINING OF RESNET-18 ON CIFAR-10

Fast networks do not always harvest the compression benefit Xu et al. (2021a); bandwidth-limited federated training is an authentic area in assessing our strategies.

**Testbed and setup.** We emulate a constrained federated learning (FL) environment with 50 CPU workers by using the same configuration as before. Additionally, we partition the CIFAR-10 dataset into 50 subsets using a Dirichlet distribution with parameter $\alpha = 10$ to mimic a non-i.i.d. data distribution among the workers. We use Top-$k$ as the base compressor in LEGACY and compare the results with no compression baseline and Accordion Agarwal et al. (2021a). This configuration more accurately reflects the limitations encountered in a real-world FL environment, characterized by heterogeneous data, constrained networks, and computational resources.

**Result.** We do not accumulate gradients at local nodes but communicate immediately to test the resilience of training when the slow network is burdened with heavy communication. Our strategies are robust in FL and outperformed the uniform Top-1.3% and Accordion, achieving a 16-35% gain in accuracy, while being $6\times$ faster than the no-compression baseline; see Figure 8b. Our layer-based policy's test accuracy is almost similar to the no-compression baseline, while the epoch-based policy outperforms the uniform Top-1.3%. The adaptive policies in LEGACY significantly lower the communicate data volume overhead in FL deployments; $B_2E_{0.6}$ and $S_{15}L_{1.2}$ communicate only 1.3% and 1.23% of the data, respectively, compared to the no-compression baseline (Figure 8c); also, see total communicated data volume during training in Figure 7c. Together, this indicates the high quality of the trained model, consistent with the findings in data center training, and validates our claim that the simple yet efficient principles in LEGACY are beneficial for federated deployments.

Table 9: CIFAR-10 experiments

| Dataset | CIFAR-10 |
|---|---|
| Architecture | AlexNet, ResNet-9 |
| Repository | Layer-Wise-AAAI20 Dutta et al. (2020) |
| | See `https://github.com/sands-lab/layer-wise-aaai20` |
| License | MIT |
| Number of workers | 2 |
| Global Batch-size | $256 \times 2$ |
| Optimizer | vanilla SGD |
| LR scheduler | piecewise-linear function that increases the learning rate from 0 to 0.4 during the first 5 epochs and then decreases to 0 till the last epoch |
| Number of Epochs | 30 |
| Repetitions | 15, with different seeds |

Table 10: Evaluation of LEGACY in large scale training of Resnet-18 on CIFAR-10.

| Methods | No compression | Top-$k$ | LEGACY-E | LEGACY-L |
|---|---|---|---|---|
| Top-1 Accuracy | 89.34 | 69.1 | 71.8 | **80.76** |

### D.3.3 SCALING LEGACY TO 100 WORKERS IN A DATA CENTER

**Testbed and setup.** We used the same configuration as in D.3.1, but scaled the system to 100 workers and removed the bandwidth limitation to accelerate training. As in the previous setup, each worker operated on an Intel Xeon Platinum 8276 CPU, and all other parameters were kept identical. The model accuracy depends on the compressed gradients being transmitted, not on their transfer speed; the results are generalizable across different bandwidths (e.g., 400 Gbps, 10 Gbps, or 1 Gbps). In practice, the compression strategy remains unchanged regardless of bandwidth; only the transfer time varies. Consequently, under low-bandwidth conditions, LEGACY provides substantial speedups over uncompressed training while also achieving higher accuracy than uniform compression.

For compression, we configured each method to transmit approximately 1% of the gradients on average, using the following parameters:

- **Top-$k$:** 1% uniform compression.

- **Top-$k$ Epoch-based:** the first epoch used a 5% ratio, the second epoch 2%, followed by 1% for the next 18 epochs (epochs 3–20). For the final 10 epochs, we applied a 0.7% ratio, resulting in an average compression ratio of 1.06%.

- **Top-$k$ Layer-based:** layers were grouped by size into four categories: small ($\leq 600$), medium ($\leq 10^5$), large ($\leq 10^6$), and very large ($\geq 10^6$). The small layer group was left uncompressed. Compression ratios of 15%, 2%, and 0.1% were applied to the medium, large, and very large groups, respectively, yielding an average compression ratio of 0.99%.

**Results and takeaways.** Table 10 reports the results for the 100-worker setting. The no-compression baseline achieves the highest accuracy (89.34%), but at the cost of substantial communication overhead. Uniform Top-$k$ suffers from a significant accuracy drop (69.1%), whereas both variants of LEGACY perform considerably better, with LEGACY-E reaching 71.8% and LEGACY-L achieving 80.76%. These results confirm that LEGACY scales effectively to 100-worker configurations while maintaining strong accuracy under aggressive compression. Combined with the earlier experiments in constrained environments, this demonstrates that LEGACY is both efficient and robust across resource-limited and large-scale deployments, making it a practical solution for challenging distributed training scenarios.

### D.4 TIME AND SPACE COMPLEXITY OF LEGACY AND OTHER ADAPTIVE COMPRESSORS

The time complexity of LEGACY is equivalent to the time complexity of the base compressor used. LEGACY does not involve any back-of-the-hand calculation in choosing the adaptive version of the compressor and needs negligible additional memory only to store the hyperparameters. Therefore, LEGACY does not require time and space complexity while granting the capability to regulate the communicated data volume, select the compressor, and decide whether to use error feedback.

Table 11: CIFAR-100 experiments

| Dataset | CIFAR-100 |
|---|---|
| Architecture | ResNet-18 |
| Repository | rethinking-sparsification Sahu et al. (2021) |
| | See `https://github.com/sands-lab/rethinking-sparsification` |
| License | MIT |
| Number of workers | 2 |
| Global Batch-size | $256 \times 2$ |
| Optimizer | SGD with Nesterov Momentum |
| Momentum | 0.9 |
| Post warmup LR | $0.1 \times 16$ |
| LR-decay | /10 at epoch 150 and 250 |
| LR-warmup | Linearly within 5 epochs, starting from 0.1 |
| Number of Epochs | 300 |
| Weight decay | $10^{-4}$ |
| Repetitions | 15, with different seeds |

Table 12: Language modelling task

| Dataset | WikiText103 |
|---|---|
| Architecture | Transformer-XL |
| Repository | NVIDIA Deep Learning Examples Nvidia |
| | See `https://github.com/NVIDIA/DeepLearningExamples` |
| License | Apache |
| Number of workers | 4 |
| Global Batch-size | 256 |
| Optimizer | LAMB |
| LR-decay | Cosine schedule from 0.01 to 0.001 |
| LR-warmup | Linearly within 1,000 iterations, reaching 0.01 |
| Number of training steps | 4500 |
| Weight decay | 0 |
| Repetitions | 4, with different seeds |

In contrast, Accordion Agarwal et al. (2021a) does not provide control over the communicated data volume since the duration of the critical regime is unknown, and it requires extra memory equivalent to twice the size of the model to store accumulated gradients from the current and previous epoch used in the algorithm. L-GreCo Markov et al. (2024) require additional computations, which are hidden during training by invoking L-GreCo infrequently (once per epoch in the conducted experiments) and also necessitates extra memory to keep accumulated gradients and some intermediate matrices used in the algorithm (according to the paper, it possesses a time complexity of $O(D|L||C|)$ and a memory complexity of $O(|L|D)$), where $|L|$ is the number of layers, $C$ a list of compression parameters to tests, and $D$ is the discretization factor (with default value $D = 10,000$)—the hyperparameters used in L-Greco. Finally, Adacomp Chen et al. (2018a), variance-based method Tsuzuku et al. (2018), and CAT Khirirat et al. (2021) do not give the possibility to select the compression method or control the communicated volume.

## E  LIMITATION AND FUTURE DIRECTION

Although adapting compression ratios based on layer size and training phase can significantly enhance model performance, it introduces additional hyperparameters. These hyperparameters define the groups and the compression ratios for each group. While following the guidelines established by `LEGACY` can improve performance, identifying optimal hyperparameters is challenging and depends on multiple factors, such as the dataset, DNN model architecture, network bandwidth, and the number and performance of workers. Regardless, in Section C.1, we presented a simplified approach that requires only two hyperparameters. In the future, we aim to develop more robust methods for selecting `LEGACY` parameters, making the process more efficient and adaptable to varying scenarios.

In this work, we explored the impact of adapting compression based on two parameters—layer size and training iteration—that require no additional computation to determine. Additionally, we investigated the potential of optimizing compression using the layer position within the model. To evaluate this, we grouped layers into two or three categories based on their position: the first and

Table 13: Recommendation task

| Dataset | Movielens-20M |
|---|---|
| Architecture | NCF |
| Repository | NVIDIA Deep Learning Examples Nvidia |
| | See https://github.com/NVIDIA/DeepLearningExamples |
| Number of workers | 2 |
| Global Batch-size | $2^{20}$ |
| Optimizer | ADAM |
| ADAM $\beta_1$ | 0.25 |
| ADAM $\beta_2$ | 0.5 |
| ADAM LR | $4.5 \times 10^{-3}$ |
| Number of Epochs | 30 |
| Weight decay | 0 |
| Dropout | 0.5 |
| Repetitions | 10, with different seeds |
| License | Apache |

Table 14: ImageNet experiments

| Dataset | ImageNet |
|---|---|
| Architecture | ResNet-50 |
| Repository | PyTorch Examples PyTorch |
| | See https://github.com/pytorch/examples |
| License | BSD 3-Clause |
| Number of workers | 4 |
| Global Batch-size | 256 |
| Optimizer | SGD |
| Momentum | 0.9 |
| LR-decay | LR decayed by 10 every 30 epochs |
| Number of Epochs | 50 |
| Weight decay | $10^{-4}$ |
| Repetitions | 1 |

last halves, or the first, middle, and last parts, respectively. Similar to our previous experiments, we applied different compression strategies (aggressive for one group and mild for another, while maintaining a similar data volume) and compared them to uniform compression. Our results did not identify a clear best approach; strategies that performed well in one case failed in another. We attribute this to model layer distributions—some models have smaller layers predominantly at the beginning, while others have them concentrated at the end. Ignoring these distributions when applying compression can lead to aggressive compression on smaller layers, resulting in degraded performance. We believe a more in-depth study that integrates layer position with layer size and training phase could refine compression parameter selection and improve performance outcomes.

The paper primarily focuses on synchronous communication and does not consider addressing asynchronous SGD setups, which is a nontrivial extension of this work. However, the dynamic compression strategies proposed by LEGACY are based on layer size and training phase, which are general principles and could be adapted for asynchronous setups. This extension would require further careful theoretical and experimental validation to ensure convergence and efficiency, and is left to future work.

LEGACY is feasible for real-time edge deployments because it uses lightweight meta-scheduling on a base compressor based on simple factors like layer size and training phase, avoiding computationally intensive methods. The framework does not rely on hard-to-compute gradient statistics, making it suitable for resource-constrained environments like edge devices. While the paper does not provide explicit latency metrics for real-time edge deployments, it emphasizes its applicability in several comparable settings: Minimal computational overhead in scheduling compression parameters; Empirical results on distributed setups (e.g., training on GPUs with 400 Gbps bandwidth) show that LEGACY does not introduce significant delays compared to uniform compression strategies. To further harvest the compression benefit of LEGACY in bandwidth-limited federated training, we deploy it using 50 and 100 workers, with every worker operating on an Intel Xeon Platinum 8276 CPU instead of a GPU, sharing a 1Gbps network bandwidth and no constraint on the bandwidth, respectively. In both cases, LEGACY demonstrates superior performance compared to the base uniform compressor. Taken together, LEGACY's lightweight nature makes it a promising candidate for edge deployments.

With the advent of large multimodal models, fine-tuning can be seen as training a DNN model with a special starting point. So, LEGACY's compression strategy, based on layer size and training phase, can be applied equally well to fine-tuning as it does to training from scratch. Fine-tuning typically involves fewer updates for certain layers, so LEGACY for fine-tuning will dynamically adjust compression parameters to these layers, making it robust to both training and fine-tuning scenarios. However, when LEGACY relies on dynamic adjustments during different training phases, in scenarios of fine-tuning with very short training durations, the benefits of LEGACY scheduling may be diminished.

Finally, we note that adversarial attacks are orthogonal to the gradient compression strategies. However, LEGACY's simple, layer- and epoch-based dynamic meta-scheduling makes it less prone to adversarial gradient perturbations compared to adaptive compression strategies that depend on gradient magnitudes or other dynamic metrics. Thus, LEGACY can be combined safely with adversarial attacks-resilient methods (methods using robust aggregation mechanisms). However, to test this efficacy is not in the primary scope of this work.

## F    Ethics statement and potential negative impact

Gradient compression techniques have been widely adopted since their introduction to the machine learning community. The strategies used in developing our adaptive compression scheduler in this work, theoretically and empirically, demonstrate their capability of achieving better accuracy in DNN training in a distributed and federated setup. The present work is theoretically driven, and experiments corroborate the theoretical claims. Therefore, we do not see any foreseeable harm it can pose to human society. However, it is always possible that some individual or an organization can use this idea to devise a *technique* that can appear harmful to society and bear evil consequences. As authors, we are absolutely against any detrimental usage, regardless of, by any individual or organization, under profit or non-profitable motivation, and pledge not to support any detrimental endeavors concerning our idea therein.

