# OpenReview forum: "LEGACY: A Lightweight Dynamic Gradient Compression Strategy for Distributed Deep Learning"
_ICLR.cc/2026/Conference — ICLR 2026 Poster_

### Official Review · Reviewer_i63J · 2025-11-01

**Soundness:** 3
**Presentation:** 3
**Contribution:** 3
**Rating:** 6
**Confidence:** 3

**Summary:**

This manuscript introduces a practically-driven compression strategy for federated learning, supported by theoretical analysis and extensive benchmarking. A central finding is that compression should be adapted to individual layer sizes and the training phase. This insight offers valuable guidance for designing more efficient communication in federated learning. The work effectively balances theory and experimentation, providing useful design principles for adaptive compression methods.

**Strengths:**

1、The paper is supported by rigorous simulations that empirically validate the proposed method. The experiments are well-constructed, employing relevant baselines and metrics to clearly demonstrate the approach's effectiveness across diverse conditions.
2、Complementing the empirical work, the authors include a theoretical analysis that grounds the core principles of their compression strategy. Although not the paper's central thrust, this analysis adds valuable depth and strengthens the overall argument.

**Weaknesses:**

1、While the paper explores a broad parameter space, the sensitivity of the method to specific parameter choices and its potential failure conditions remain unclear. A more systematic investigation, or at least a discussion of possible failure modes, would help to better assess the approach's reliability and practical applicability.
2、Regarding the theoretical contribution: as noted in the manuscript, the analysis of compression operators and their convergence rates appears to align with existing theoretical frameworks. It would be helpful to clarify the novel theoretical insights offered by this work compared to prior analyses.
3、The models used in the experiments are relatively small and may not reflect modern large-scale architectures. Since their communication overhead is inherently limited, the results may not fully demonstrate the method's effectiveness in high-communication settings. Experiments on larger generative models, such as GPT-family architectures, would strengthen the evaluation.

**Questions:**

1、To better assess the practical applicability of the method, could the authors delineate the scenarios or conditions under which the proposed compression framework might exhibit performance degradation? A discussion of such limitations would be valuable for understanding the robustness of the approach.
2、The theoretical analysis presented, while sound, appears to align with established convergence rates for standard compressors in the literature. Could the authors clarify the novel theoretical contribution of their work in this context?
3、The experimental validation primarily uses smaller-scale architectures. To demonstrate the scalability and general applicability of the method, could the authors provide results on larger, contemporary models, such as generative transformers (e.g., GPT variants)?

---

> ### Author Response · Authors · 2025-11-25
> **Rebuttal**
>
> We thank the reviewer for their feedback and appreciate the positive evaluation of the soundness, presentation, and contribution of our work. Below, we provide our responses to the questions.
>
> **W.1 & Q.1: Limitation.**
>
> Please see the Limitations and Future Directions in Section E. We also note that certain areas remain unexplored in our current work, including scenarios such as adversarial training, asynchronous communication, and settings where workers have varying communication budgets. These conditions may affect the performance of LEGACY, and further theoretical and empirical investigation is needed to fully understand its behaviour in such environments.
>
> **W.2 & Q.2: Theoretical contribution.**
>
> We thank the reviewer for the question. We believe the comment “as noted in the manuscript, the analysis of compression operators and their convergence rates appears to align with existing theoretical frameworks.” refers to the sentence before Theorem 1, which states: “the following theorem gives the complexity results for unbiased $\delta_t$-compressors, which are similar to the classical complexity results for compressed SGD-type algorithms; see Dutta et al. (2020); Stich & Karimireddy (2020); Sahu et al. (2021)”. To clarify, these cited works are not on adaptive compression methods; they discuss the convergence of compressed distributed first-order algorithm (SGD) under a nonconvex loss function for different strategies, such as threshold sparsification, layer-wise vs. full-model compression, and error feedback. These premises are similar to ours, and we cite them to position our analysis within the broader theory of compressed communication in a distributed setting. A first-order algorithm, with or without compression, will have the same asymptotic convergence rate with a variance term associated with it, which is governed by different factors; please see Lines 313-350. In case of LEGACY, this term is affected by a varying compression ratio, $\delta_t$ at each iteration, which is due to LEGACY’s layer- or training epoch-based strategies.
>
> More precisely, our analysis in Section 3 is self-contained and specifically adapted to the LEGACY framework. It handles a general $\delta_t$-compressor at each iteration and establishes **nonconvex convergence of compressed distributed SGD** for both biased and unbiased compressors; see Theorem 1. Additionally, we note that most adaptive gradient compression methods provide **no theoretical guarantees.** Among the five adaptive baselines we evaluate (CAT, Variance-based compression, Accordion, AdaComp, and L-Greco), only CAT includes theoretical results, and only in the **single-node** setting.
>
> In contrast, our results establish convergence for general compressors without such restrictions, offering a more practical and broadly applicable foundation for dynamic compression and distinguishing LEGACY from heuristic or compute-heavy adaptive approaches.
>
> **W.3 & Q3: Experiments on larger generative models.**
>
> We thank the reviewer for this valuable suggestion. **We used six DNN models and five datasets in our experiments, including ResNet-50, ResNet-18, Transformer-XL, ImageNet-1K, and WikiText.** These are widely used benchmark models and datasets in the gradient compression literature, which allow for fair and direct comparisons with prior work.
>
> That said, we agree that evaluating LEGACY on more modern architectures and datasets (such as GPT variants) would further demonstrate its generality. Following the reviewer’s recommendation, we expanded our evaluation by conducting additional **GPT-2 small (124M parameters) training experiments on the OpenWebText corpus ($\approx 40$ GB)**. The setup and results have been added to Section 4.4 of the main paper. These experiments show that LEGACY remains effective on modern transformer workloads: both variants consistently outperform the uniform Top-k baseline while communicating a comparable amount of data, and the layer-based variant achieves a notably lower validation loss (a 4.14% improvement).

---

### Official Review · Reviewer_NtMi · 2025-11-02

**Soundness:** 2
**Presentation:** 3
**Contribution:** 2
**Rating:** 6
**Confidence:** 3

**Summary:**

The paper introduces LEGACY, a simple scheduler to make gradient compression dynamic in distributed deep learning. Instead of using a fixed compression ratio, LEGACY adapts compression by training phase (light compression early, heavy later) and by layer size (compress large layers more, small layers less).
The scheduler works with any existing compressor without additional compute overhead or per-iteration tuning. Theoretical analysis shows the intuition behind the proposed method and guarantees the convergence of compressed stochastic gradient descent. Comprehensive experiments under different settings demonstrate the effectiveness of LEGACY compared to other compression methods.

**Strengths:**

1. The paper explores a dynamic compression strategy that adapts to training phase and layer size, which can potentially improve the efficiency of distributed deep learning.
2. It provides sufficient experimental results under various settings that support the effectiveness of the proposed method.
3. The proposed compression method is lightweight and can be integrated with other compression techniques.

**Weaknesses:**

Compared to existing adaptive compression methods such as Accordion and L-GreCo (Section 4.3.1), LEGACY does not clearly demonstrate advantages in either accuracy or data volume. Therefore, the novelty and practical impact of the method appear somewhat limited.

**Questions:**

1. In practice, is the chosen compression schedule consistent with the theoretical assumptions and requirements?  There appears to be a gap between the theory and the experimental settings that is not clearly explained or discussed.
2. The paper introduces two dynamic compression strategies: by training phase and by layer size. And it also states that they can mix the two strategies up. Is there any theoretical analysis for the mixed strategy?

---

> ### Author Response · Authors · 2025-11-25
> **Rebuttal**
>
> We thank the reviewer for an overall positive evaluation of our paper. Below, please find our answers to the questions.
>
> **W1. Limited advantages compared to other approaches**
>
> We thank the reviewer for this observation. Upon revisiting the L-GreCo public implementation, we discovered that PowerSGD, Accordion, and L-GreCo disable compression during the first 1,000 iterations, and our initially reported compression ratios were computed only over the iterations where compression was active. After recalculating the averages over the full training run, the advantage of LEGACY becomes clearer: PowerSGD, L-GreCo, and Accordion each transmit more than 2% of the total data volume for 74.58%, 75.23%, and 75.11% accuracy, respectively. In contrast, **LEGACY-L and -E communicate only about 1.1% of the total volume while achieving 75.21% and 75.55% accuracy.** Thus, **LEGACY attains higher accuracy while transmitting roughly half the communication volume** of L-GreCo and Accordion.
> Additionally LEGACY main contribution lies in its lightweight, general, and theoretically grounded design. Unlike other adaptive compressors, LEGACY introduces no additional memory or computational overhead. For instance, L-GreCo has a time complexity of $O(D|L||C|)$ and a memory complexity of $O(|L|D)$, where $|L|$ is the number of layers, $C$ is a list of compression parameters to test, and $D$ is the discretization factor. In contrast, LEGACY requires only a simple lookup to determine the compression factor based on layer size or training phase. Moreover, LEGACY is compressor-agnostic and can operate with or without error feedback, providing users explicit control over the communication budget an option unavailable in many existing methods.
>
>
> In addition, LEGACY employs intuitive and easily interpretable hyperparameters. Thresholds are used to group layers by size or divide training into phases, and the compression parameter list directly specifies the compression level per group or phase, offering users a clear understanding of how each setting impacts communication volume. In contrast, many adaptive approaches (e.g., Accordion, L-GreCo, and variance-based compression methods) **rely on non-trivial hyperparameters**, such as $\zeta$, $\alpha$, and $\eta$ thresholds, discretization factors, or execution frequencies. They lack a transparent relationship to the resulting compression ratio and often depend heavily on the model architecture or dataset. These thresholds typically control compression based on gradient values or other statistics, which can be hard to obtain and **can make the resulting communication volume vary even across different random seeds.**
>
> In summary, LEGACY provides a theoretically justified, numerically validated, and lightweight framework that is simpler to tune and more broadly applicable than existing adaptive compression methods.
>
> **Q.1 & Q.2 Consistency with theoretical assumptions and analysis of mixed strategy.**
>
> We thank the reviewer for the questions. Our theoretical analysis in Section 3 is built on classical assumptions that are standard in the gradient-compression literature; in Section B.1, we also cite prior works that rely on these same assumptions. Importantly, the convergence theorem is derived for a general adaptive $\delta_t$-compressor, without imposing any specific structure on how $\delta_t$ varies over iterations or across layers. The theory only requires that, at each iteration, the compressor satisfies the $\delta_t$-compression property; it does not prescribe the exact schedule or functional form of $\delta_t$.
>
> Because of this generality, the compression schedules used in practice, whether based on training phase, layer size, or a combination of both, are fully consistent with our theoretical framework. For the mixed strategy, combining the two rules simply produces a $\delta_t$ value for each layer at each iteration. This remains a valid $\delta_t$-compressor sequence and is therefore directly covered by the theoretical analysis. No separate theorem is needed; the convergence guarantees apply unchanged to any such mixed strategy. We will add this remark after Theorem 1, our nonconvex convergence result for unbiased compressors.

---

### Official Review · Reviewer_xMGk · 2025-11-04

**Soundness:** 3
**Presentation:** 2
**Contribution:** 3
**Rating:** 6
**Confidence:** 3

**Summary:**

This paper presents a communication-efficient federated learning method. The proposed method is scheduler of gradient compression, which is agnostic to compression (quantization) method. The proposed method is based on the observation that the number of parameters for individual layers could be very different. Furthermore, at different stages of model training, different compression strength is necessary to achieve good balance between performance and communication cost reduction. To demonstrate the above ideas, this paper provides theoretical analysis on the compression bounds and convergence. Although theoretical analysis is provided, the experiments are still empirical. Testing on several datasets and model architectures, including ResNet and Transformer, this paper shows better results with different quantization methods, compared with other adaptive compression techniques like Accordion.

From my perspective, the complexity of layers and the training stages are essentially related to gradient norm (later epochs produce smaller norms) and direction. It can be analyzed from the compression errors in norm and cosine similarity after applying the quantization. Therefore, a more controlled scheduler might be easier to deploy than an empirical method studied in this work, by checking the compression errors caused by layer size and training epoch numbers.

I vote this paper as "6: marginally above the acceptance threshold. But would not mind if paper is rejected", but could change my decision in my final voting.

**Strengths:**

1. A very simple and quantization-agnostic approach is proposed for communication-efficient federated learning. Based on this approach, several models and datasets are applied with better performance than previous work.

2. Theoretical analysis is provided, which enhances the motivation of this work.

**Weaknesses:**

1. The selection of compression strength is still empirical. It increases the challenge of deployment of this model in the real world. For example, the data from individual clients might be very different, that some clients may provide training data very different to major distribution. For this client, the gradient norm will be very different to others. Therefore, different clients may also need different compression strength.

2. It will be helpful to show the performance curves over different training epochs and different communication costs.

**Questions:**

In the research history, it's known that early layers are learned at the beginning of training. In particular, when we finetune from a pretrained model, the early layers do not change drastically, compared with later layers. It is interesting to know how to balance the compression strength for lower / higher layers at the beginning / end of training.

---

> ### Author Response · Authors · 2025-11-25
> **Rebuttal**
>
> We thank the reviewer for the constructive and positive evaluation, as well as for the valuable suggestion on incorporating layer position into the compression design. We hope the responses below address the concerns raised and justify a higher score.
>
> **W1. Non-i.i.d. data distribution.**
>
> We thank the reviewer for the insightful comment. We evaluated LEGACY under **non-i.i.d. data distribution and CPU-based training** settings to simulate heterogeneous and resource-limited environments. Due to space constraints, these results are reported in **Appendix D.3.2.** Regarding device heterogeneity, LEGACY is lightweight and introduces minimal compute overhead—runtime is primarily determined by gradient computation and the base compressor rather than the scheduling strategy itself (see **Appendix D.4 for further discussion**).
>
> We acknowledge that clients with different communication budgets are an important direction and plan to explore such settings if time permits. Nonetheless, we expect LEGACY to maintain strong performance since its core motivation still holds---slightly increasing compression aggressiveness on large layers allows significant relaxation of compression on smaller ones, while gradients naturally become smaller as training progresses—making later epochs more tolerant to compression.
>
> **W2. Performance curves.**
>
> We thank the reviewer for this remark. We intentionally avoided plotting performance curves over epochs because we evaluated many compression configurations; see Figure 2. For each average communication ratio, we report the uniform baseline, two LEGACY variants, and the two inverse approaches in Figure 2. Including all corresponding performance curves would overcrowd the plots and reduce readability. We humbly note that this is a standard practice in the literature. E.g., please see [1], and we followed this practice. However, as the reviewer pointed out, we provide the training performance plots in our CPU experiments and in the comparisons with other adaptive approaches; please see Figures 4, 7, and 8.
>
> **Q1. Incorporating layer position.**
>
> **Answer.** We thank the reviewer for highlighting this interesting point. As discussed in **Appendix E (Limitations)**, we explored varying the compression aggressiveness based on layer position (e.g., early, middle, and later layers) by testing different configurations—such as increasing compression on the final layers to allocate more budget to the initial ones, and vice versa. However, we did not observe a consistent winning strategy. That is, configurations that improved performance on some models failed on others. We attribute this to differences in layer size distributions across models; in some architectures, small layers dominate certain regions, and increasing compression aggressiveness in these regions can overly compress sensitive layers. While incorporating layer position into compression control is indeed an intriguing direction, it likely requires jointly considering factors such as layer size and training phase. We therefore leave this as a promising extension for future work.
>
>
> **Reference**
>
> [1] Hang Xu, Chen-Yu Ho, Ahmed M. Abdelmoniem, Aritra Dutta, El Houcine Bergou, Konstantinos Karatsenidis, Marco Canini, and Panos Kalnis. Grace: A compressed communication framework for distributed machine learning. In IEEE ICDCS, 2021a.

---

### Official Review · Reviewer_WieB · 2025-11-04

**Soundness:** 3
**Presentation:** 2
**Contribution:** 3
**Rating:** 4
**Confidence:** 4

**Summary:**

This work itroduce  a lightweight, open-source framework, referred to as LACACY, which dynamically compresses gradients in distributed DNN training. Its scheduler, guided only by layer size and training phase, offers a simple  alternative to compute-intensive adaptive compressors. Benchmarking with Top-k, Random-k, QSGD, and PowerSGD shows that LEGACY achieves consistent performance gains over uniform on Cifar10, Cifar100 and other datasets.

**Strengths:**

The paper presents a unified framework that enables various communication-compression algorithms to dynamically adjust their compression levels. Experimental results demonstrate that this proposed scheme is both simple and effective.

**Weaknesses:**

1. **Theoretical Motivation:** The motivations for the dynamic compression strategies are not fully convincing.
   * The **training phase-based** strategy is justified by Lemma 1, which relies on convergence properties of Gradient Descent for convex objectives. However, this theoretical foundation may not directly translate to the non-convex and stochastic optimization of Deep Neural Networks, limiting its applicability.
   * The **layer size-based** strategy relies on the unverified assumption that larger layers are more over-parameterized and can tolerate higher compression. This claim lacks theoretical or empirical support within the paper.

2. **Insufficient Experimental Evidence:** While the paper claims LEGACY is a "lightweight yet effective dynamic scheduler," the experiments do not fully substantiate this claim.
   * There is no evidence demonstrating that LEGACY's lightweight design translates to significant wall-clock time reduction compared to compute-intensive adaptive methods like L-GreCo and Accordion. The time saved by its simple scheduler might be negligible compared to overall compression computation.
   * Performance comparisons against key counterparts are limited, primarily appearing in the ResNet-18 on CIFAR-100 experiment. This narrow scope limits generalizability.
   * The reported accuracy for uniform PowerSGD and L-GreCo is notably lower than results in the original L-GreCo paper [1] under similar settings. Additionally, the compression ratio used for Accordion is lower than for other methods, potentially creating an unfair comparison. These issues undermine the reliability of the experimental findings.
In summary, the comparative experiments lack breadth, and their results are not entirely convincing.


[1] Alimohammadi, Mohammadreza, et al. "L-GreCo: Layerwise-Adaptive Gradient Compression for Efficient and Accurate Deep Learning." arXiv:2210.17357, 2022.

**Questions:**

1. The architectures (e.g., ResNet-9, ResNet-18) and datasets (e.g., CIFAR-10/100, WikiText) used for evaluation are somewhat outdated. Could the authors demonstrate LEGACY's effectiveness on more modern benchmarks (e.g., Vision Transformers or contemporary datasets)?

2. In Table 5, the reported baseline accuracy for training ResNet-50 on ImageNet-1K is 59.43%, significantly lower than standard benchmarks. Could the authors explain this discrepancy in their training setup?

---

> ### Author Response · Authors · 2025-11-25
> **Rebuttal --1**
>
> We thank the reviewer for their comments and questions. Below, we provide our responses to the points raised.
>
> **W1. Theoretical Motivation**
>
> We appreciate the reviewer’s concern. The motivation for LEGACY is both empirical and theoretical.  We begin by demonstrating in Section 1 how the behaviour of the Threshold compressor naturally adapts its compression aggressiveness with respect to both the training phase and the layer size. This empirical observation directly inspires our two dynamic strategies.
>
> The **training-phase–based** strategy is theoretically supported under gradient descent settings: Lemma 1 for the strongly convex case and Lemma 2 for the smooth non-convex case under the PL condition. While these settings are simplified, they provide principled intuition for how compression should evolve during training and are consistent with LEGACY’s empirical behaviour in distributed settings.
>
> The **layer-size–based** strategy is empirically motivated: reducing the communication budget of large layers—while reallocating the saved bandwidth to smaller layers—has a negligible effect on the large layers but yields substantial benefits for smaller ones. For example, instead of applying uniform compression of 1%, compressing the largest 25% of layers by 0.99% (which empirically performs equivalently) allows us to transmit 100% of the smallest 25% of layers in ResNet-18, producing a meaningful accuracy gain (see Subsection C.1.3 for further discussion by considering the communicated data volume).
>
> Taken together, Lemmas 1 and 2 formalize why layer size and training phase are critical factors for gradient compression. They show that in the early training stage, the gradients have high magnitudes, thus they require mild compression to preserve the convergence. In the late phase of training, aggressive compression is tolerable as gradients diminish. This part of the theory motivates LEGACY’s simplicity (no need for complex adaptations). Therefore, the reviewer’s assertion that `` this theoretical foundation may not directly translate to the non-convex and stochastic optimization of Deep Neural Networks, limiting its applicability,” is not correct.
>
> Finally, our Theoretical analysis in Section 3 makes the paper self-contained and provides **convergence guarantees for both biased and unbiased $\delta_t$ compressors under non-convex distributed SGD.** This is the practical setup as the reviewer was referring to.* Under the usual assumptions on the nonconvex loss function, $F$, for stochastic first-order algorithms in the compressed, distributed setup, we validate the influence of our policies on the convergence of compressed distributed SGD using biased and unbiased $\delta$-compressors and show that it has the same asymptotic convergence rate as the no-compression baseline. This primarily guarantees that the (nonconvex) convergence of distributed SGD with $\delta_t$-compressors in each iteration $t$ (which the policies of LEGACY govern, a changing compression level over iteration, and $\delta_t$ modulates the effect of epoch- or layer-based strategies). This is the overarching goal of the convergence result.
>
> We finally note that most existing works in adaptive gradient compression do not provide theoretical guarantees. E.g., among the five adaptive gradient compression methods (CAT, Variance-based compression, Accordion, AdaComp, and L-Greco) compared in our paper, only CAT includes theoretical results, and those are limited to the single-node setting. While theoretical analyses do exist in other works, many rely on error-feedback mechanisms, fixed compressors, or restrictive assumptions that are difficult to generalize. In contrast, our analysis establishes convergence for both biased and unbiased compressors without relying on such restrictive assumptions, providing a more general and practical foundation for adaptive compression strategies and ensuring that LEGACY achieves convergence comparable to state-of-the-art methods, distinguishing it from heuristic or compute-heavy adaptive approaches.

---

> ### Author Response · Authors · 2025-11-25
> **Rebuttal--1 (Continued)**
>
> **W2. Insufficient Experimental Evidence**
>
> **On wall-clock time.** Although we provide a wall-clock time comparison of LEGACY with Top-k and three other adaptive compressors (ADACOMP, CAT, and the Variance-Based compressor) when training ResNet-9 on CIFAR-10 in Figure 4, we appreciate the reviewer’s thoroughness. The computational advantage of LEGACY compared to L-GreCo and Accordion is indeed less apparent in our current setup due to three factors: (1) the use of high-performance GPUs (A100), (2) the fact that L-GreCo is invoked only once per epoch to reduce overhead, and (3) the use of communication–computation overlap. The difference becomes clearer when comparing their **time and memory complexities, as discussed in Section D.4.** According to the original paper, L-GreCo has a **time complexity of $O(D|L||C|)$ and a memory complexity of $O(|L|D)$**.  To operate, L-GreCo must compute the compression error and compressed size for every parameter across all possible compression parameters in the list $C$ for every layer. Consequently, when the base compressor is slow or the candidate list $C$ is large, the computational and time overhead increase significantly. In our experiments, L-GreCo was invoked only once per epoch, the set $C$ contained six candidate ranks, and the base compressor was PowerSGD, which is relatively fast. In contrast, LEGACY requires only a simple lookup of the compression degree based on either the layer size or the training phase.
>
> **On the fairness of comparisons.** For the L-GreCo experiment, we used the official implementation released by the authors. The difference in accuracy we observe compared to the original L-GreCo paper is likely due to the difference in the number of workers: our setup uses 2 workers, whereas the original evaluation used 8.
>
> We thank the reviewer for raising the fairness concern. After re-examining the public L-GreCo implementation, we found that PowerSGD, Accordion, and L-GreCo disable compression during the first 1,000 iterations to improve accuracy. Our initially reported compression ratios were computed only over the iterations where compression was active, which led to unfair comparisons. We have now recalculated all averages over the full training run.
>
> **With the corrected measurements, the conclusions become even stronger for LEGACY.** Uniform PowerSGD transmits 2.85% of the total data volume for 74.58% top-1 accuracy; L-GreCo transmits 2.19% for 75.23% top-1 accuracy; and Accordion transmits 2.02% for 75.11% top-1 accuracy. In contrast, **LEGACY-L and -E transmit only about 1.1% of the total volume while achieving 75.21% and 75.55% accuracy, respectively.**
>
> Thus, after correction, **our method not only achieves higher accuracy but also communicates approximately half the volume of L-GreCo and Accordion.**
>
>
> **Q1. Evaluation of modern architectures**
>
> **Answer.** We agree with the reviewer that testing LEGACY on newer models would further demonstrate its generality. While such large-scale tests could enrich the paper, our current results already show strong performance and scalability, with model diversity and system scale comparable to or better than the existing literature. We note that the datasets and models used in the paper (e.g., ResNet-50/ImageNet-1K, ResNet-18, CIFAR-10/100, Transformer-XL/WikiText) are among the most used benchmarks in the gradient compression and distributed training literature, enabling fair comparison with prior work.
>
> We acknowledge that evaluating LEGACY on more modern architectures and datasets would better showcase its generality. In response to the reviewer’s suggestion, we conducted additional experiments by **training GPT-2 on the OpenWebText dataset.** These results confirm that LEGACY continues to perform well on modern transformer workloads: both variants consistently outperform the uniform Top-k baseline while communicating a comparable amount of data, and the layer-based version achieves a notably lower validation loss (4.14% improvement).
>
> **Q2. Low baseline accuracy for ImageNet-1K.**
>
> **Answer.** We thank the reviewer for the observation. The lower baseline accuracy reported for ResNet-50 primarily results from computational and time constraints, given the extensive range of compression parameters evaluated in our study and the large size of the ImageNet dataset, which makes full training time-consuming. To ensure a feasible experimental setup, we trained ResNet-50 for 50 epochs, whereas the standard benchmark typically uses 90 epochs in the original implementation. This shorter training schedule accounts for the observed difference in baseline accuracy. We note that such a setup is common practice in academic evaluations, and our experiments include both the no-compression baseline and the uniform compression baseline as references for fair comparison.

---

### Author Response · Authors · 2025-12-02
**Authors' Final Remarks**

Dear AC and SAC,

We thank the reviewers for their constructive feedback on our work and for recognizing the potential and novelty of our work. No reviewer raised concerns regarding the clarity, novelty, or theoretical soundness of the paper. The comments were focused on scalability and applicability of LEGACY to larger models, such as GPT. **In addition to evaluating LEGACY on classical benchmarks, which include ResNet-9, ResNet-50, and Transformer-XL models trained on CIFAR-100, ImageNet-1K, and WikiText-103, we further evaluate LEGACY in a modern setting by training GPT-2 small (124M parameters) on the OpenWebText corpus ($\sim40$ GB).** Below, we summarize the key clarifications for the Area Chair.

(1) **Experiments on GPT (Reviewers WieB, i63J)**. We added new evaluations on GPT-2 trained on the OpenWebText dataset, demonstrating that LEGACY retains its effectiveness in contemporary transformer-based workloads. *Both LEGACY variants outperform the uniform Top-k while transmitting a similar amount of data, with the layer-based strategy yielding a 4.14% lower validation loss relative to uniform Top-k.*

(2) **Significance of comparisons with L-GreCo and Accordion (Reviewers WieB, NtMi).** Upon re-examining the public implementation, we discovered that PowerSGD, Accordion, and L-GreCo apply no compression during the first 1,000 iterations, whereas our initial average compression calculations only accounted for iterations where compression was executed. *After correcting this and recomputing the data volume, the advantage of LEGACY becomes clearer, and we thank the reviewers for this.* LEGACY achieves a similar or slightly better accuracy while communicating roughly half the data. Uniform rank-3 PowerSGD transmits about **2.85% of the total data volume and achieves 74.58% test accuracy, L-GreCo transmits about 2.19% of the total data volume and achieves 75.23% test accuracy, Accordion transmits about 2.02% of the total data volume and achieves 75.11% test accuracy.** In contrast, LEGACY-L and -E transmit about **1.1% of the total data volume and achieve 75.21% and 75.55% test accuracy, respectively.** We revised the paper accordingly.

(3) **Clarifying theoretical motivation (Reviewers WieB, NtMi, i63J).** While none of the reviewers questioned the correctness of our theory, some reviewers asked for further clarification on our analysis. We emphasize that the analysis in Section 2 directly motivates and validates the design of LEGACY. The nonconvex convergence analysis in Section 3 provides a unifying and self-contained theoretical foundation for an adaptive $\delta_t$ gradient compressor (which the policies of LEGACY govern---a changing compression level over iteration and $\delta_t$ modulates the effect of epoch- or layer-based strategies), both biased and unbiased, in a distributed setting using $n$ nodes under a master-worker communication protocol. **We clarify that theoretical analysis is still largely lacking in adaptive compression. Among the five adaptive compressors compared in our study, only CAT provides theoretical guarantees, and only in a single-node setting.**

We believe that we have fully addressed the reviewers’ concerns by (1) adding GPT-2 experiments, (2) correcting and clarifying the L-GreCo comparison, and (3) explaining the role and novelty of our theoretical contributions. **We also note that Reviewer xMGk explicitly indicated openness to improving their score.**

The paper, including the appendix, is revised and updated. The new additions and modifications are done in **text color Blue**. The deletions are highlighted using **Blue Strikethrough**. We modified the **Abstract,** **Section 1 Introduction**, **Section 2 (Lines 192-193),** and **Section 4**. We added a completely new **Section 4.4 Evaluating LEGACY on GPT-2** and **Table 2**. We also modified **Section 4.3: Power SGD experiment details and results** and **Table 3** to include the correct data volume for L-GreCo, Power-SGD, and Accordion.

In summary, we present a lightweight, theoretically grounded, and practically effective adaptive compression framework, validated across diverse compressors, models, and datasets. We respectfully ask the Area Chair to consider the novelty, rigor, and completeness of the work, especially given that no reviewer identified technical errors, theoretical flaws, or substantive limitations.

Sincerely,

The Authors of LEGACY

---

### Meta-Review · Area_Chair_TkCv · 2026-01-04

**Summary:**

This paper develops two new compression strategies to reduce communication costs in distributed learning, based on the training stage and the layer size. It provides both theoretical analysis and empirical evaluation.

All reviewers acknowledge that the proposed strategies are interesting and effective. The main concerns relate to the limited scope of the evaluation. In the rebuttal, the authors provided additional empirical results on larger models, and these results further confirm the effectiveness of the proposed methods.

However, the paper still has two limitations. First, the selection or combination of the two strategies remains heuristic, and the theoretical analysis does not distinguish their relative performance. Second, the theoretical results rely on some uncommon or strong assumptions, such as Assumptions 8 and 9. Establishing convergence rates under milder assumptions would strengthen the paper.

Overall, the proposed methods are interesting and achieve superior empirical performance. Considering the demonstrated improvements, I recommend acceptance.

**Reviewer Concerns:**

A major concern was the evaluation on large models. The authors have provided new results on a large GPT model, and the performance confirms the effectiveness of the proposed method.

The concern regarding the theoretical analysis of the combination of the two strategies remains unresolved. However, this issue is challenging and does not significantly weaken the contribution of the paper.

**Reviewer Scores:**

Three positive reviewers will remain positive.

The concerns of the negative reviewers, mainly about empirical evaluation, have been addressed by more explanation and new results. Therefore, this reviewer might raise the initial score.

---

### Decision · Program_Chairs · 2026-01-26

Accept (Poster)